# Association of socioeconomic deprivation with asthma care, outcomes, and deaths in Wales: A 5-year national linked primary and secondary care cohort study

Mohammad A. Alsallakh[1,2,3]*, Sarah E. Rodgers[4], Ronan A. Lyons[1,2], Aziz Sheikh[2,3,5], Gwyneth A. Davies[1,3]

1 Swansea University Medical School, Swansea, United Kingdom, 2 Health Data Research UK, Swansea and Edinburgh, United Kingdom, 3 Asthma UK Centre for Applied Research, Edinburgh, United Kingdom, 4 Department of Public Health and Policy, University of Liverpool, Liverpool, United Kingdom, 5 Usher Institute, The University of Edinburgh, Edinburgh, United Kingdom

* M.A.AlSallakh@swansea.ac.uk

**Data Availability Statement:** This study makes use of the following linked anonymised core

## Abstract

### Background

Socioeconomic deprivation is known to be associated with worse outcomes in asthma, but there is a lack of population-based evidence of its impact across all stages of patient care. We investigated the association of socioeconomic deprivation with asthma-related care and outcomes across primary and secondary care and with asthma-related death in Wales.

### Methods and findings

We constructed a national cohort, identified from 76% (2.4 million) of the Welsh population, of continuously treated asthma patients between 2013 and 2017 using anonymised, person-level, linked, routinely collected primary and secondary care data in the Secure Anonymised Information Linkage (SAIL) Databank. We investigated the association between asthma-related health service utilisation, prescribing, and deaths with the 2011 Welsh Index of Multiple Deprivation (WIMD) and its domains. We studied 106,926 patients (534,630 person-years), 56.3% were female, with mean age of 47.5 years (SD = 20.3). Compared to the least deprived patients, the most deprived patients had slightly fewer total asthma-related primary care consultations per patient (incidence rate ratio [IRR] = 0.98, 95% CI 0.97–0.99, p-value < 0.001), slightly fewer routine asthma reviews (IRR = 0.98, 0.97–0.99, p-value < 0.001), lower controller-to-total asthma medication ratios (AMRs; 0.50 versus 0.56, p-value < 0.001), more asthma-related accident and emergency (A&E) attendances (IRR = 1.27, 1.10–1.46, p-value = 0.001), more asthma emergency admissions (IRR = 1.56, 1.39–1.76, p-value < 0.001), longer asthma-related hospital stay (IRR = 1.64, 1.39–1.94, p-value < 0.001), and were at higher risk of asthma-related death (risk ratio of deaths with any mention of asthma 1.56, 1.18–2.07, p-value = 0.002). Study limitations include the deprivation index being area based and the potential for residual confounders and mediators.

datasets held in the Secure Anonymised Information Linkage (SAIL) Databank at Swansea University, Swansea, UK: Welsh Demographic Service Dataset (WDSD) Welsh Longitudinal General Practice (WLGP) dataset Emergency Department Data Set (EDDS) for Wales Patient Episode Database for Wales (PEDW) Annual District Death Extract (ADDE) dataset We would like to acknowledge all the data providers who make anonymised data available for research. The anonymised person-level data used in this study are held by SAIL, and cannot be shared publicly. All applications to access SAIL data can be made at https://saildatabank.com/application-process/ and are carefully reviewed by an independent Information Governance Review Panel (IGRP) to ensure proper and appropriate use of data. When approved, access is then provided through the SAIL Gateway, a privacy-protecting safe haven and a secure remote access system.

**Funding:** This work was funded by Health and Care Research Wales (https://www.healthandcareresearch.gov.wales) and Swansea Bay University Health Board (https://sbuhb.nhs.wales) (GAD, RAL, SER). We acknowledge the support of the Asthma UK Centre for Applied Research (AUKCAR) and Health Data Research UK. We also acknowledge the support of BREATHE - The Health Data Research Hub for Respiratory Health (MC_PC_19004), which is funded through the UK Research and Innovation Industrial Strategy Challenge Fund and delivered through Health Data Research UK. GAD receives a salary from Swansea Bay University Health Board for her honorary respiratory consultant post. The funders had no role in study design, data collection and analysis, decision to publish, or preparation of the manuscript.

**Competing interests:** I have read the journal's policy and the authors of this manuscript have the following competing interests: RAL is supported by Health Data Research UK (HDR-9006), which is funded by the UK Medical Research Council, Engineering and Physical Sciences Research Council, Economic and Social Research Council, National Institute for Health Research (England), Chief Scientist Office of the Scottish Government Health and Social Care Directorates, Health and Social Care Research and Development Division (Welsh Government), Public Health Agency (Northern Ireland), British Heart Foundation and Wellcome. AS is an Academic Editor on PLOS Medicine's editorial board and declares support from the Asthma UK Centre for Applied Research.

**Abbreviations:** A&E, accident and emergency; AMR, asthma medication ratio; BTS/SIGN, British

## Conclusions

In this study, we observed that the most deprived asthma patients in Wales had different prescribing patterns, more A&E attendances, more emergency hospital admissions, and substantially higher risk of death. Interventions specifically designed to improve treatment and outcomes for these disadvantaged groups are urgently needed.

## Author summary

### Why was this study done?

- Income, education, and region of living are known to affect a person's health, and studies around the world found links between asthma and these socioeconomic factors.

- However, little is known about how the different types of socioeconomic disadvantage affect asthma across the lifetime.

### What did the researchers do and find?

- We studied 106,926 people with treated asthma in Wales for 5 years and used an official metric to rank areas of residence by wealth, education, and other factors.

- We analysed the links between this metric and how often people with asthma go to general practitioners (GP), receive medications, or suffered severe asthma attacks. We also analysed the link with asthma death in 327,906 people with asthma.

- We found worse asthma outcomes in the more disadvantaged areas, especially those with lower levels of wealth, employment, and education.

- In the most disadvantaged areas, people went more often to emergency departments for asthma, were approximately 50% more likely to be admitted to hospital and die from asthma, had a slightly worse balance of asthma medications with lower ratios of controller-to-total asthma medications, and were 3 times more likely to take 12 or more reliever inhalers per year compared to people in the least disadvantaged areas.

### What do these findings mean?

- People with asthma in the more disadvantaged areas have worse control of the disease, experience more asthma attacks, and are at higher risk of death from asthma.

- Lack of educational opportunities likely affects how well people manage their asthma and put them at higher risk of asthma attacks and death.

- The socioeconomic gap in asthma could be mitigated with GP encouragement of people to receive and take enough preventing medications and self-manage their asthma well regardless of background, and with wider policies to provide equal educational opportunities across society.

Thoracic Society/the Scottish Intercollegiate Guidelines Network; COPD, chronic obstructive pulmonary disease; fREML, fast restricted maximum likelihood; GAM, generalised additive model; GP, general practitioner; ICD-10, 10th Revision of the International Classification of Diseases; ICS, inhaled corticosteroids; IRR, incidence rate ratio; LABA, long-acting beta-adrenoceptor agonist; LOS, length of stay; LSOA, Lower layer Super Output Area; NB, negative binomial; ONS, Office for National Statistics; Q–Q, quantile–quantile; RECORD, REporting of studies Conducted using Observational Routinely-collected health Data; SABA, short-acting beta-adrenoceptor agonist; SAIL, Secure Anonymised Information Linkage; STROBE, Strengthening the Reporting of Observational Studies in Epidemiology; WIMD, Welsh Index of Multiple Deprivation; WLGP, Welsh Longitudinal General Practice.

## Introduction

Asthma is one of the most prevalent chronic diseases and has significant clinical and economic burden [1]. However, asthma burden is not evenly distributed within populations, and socioeconomic variations in asthma prevalence, emergency hospital admissions, and mortality have been recorded worldwide [2–4]. These variations have been attributed to a range of modifiable and non-modifiable factors. Ethnicity has been found to partly explain the higher asthma prevalence, disease severity, and risk of asthma admissions among South Asians and Afro-Caribbeans in the United Kingdom [5,6] and African-Americans and Puerto Ricans in the United States [7]. However, lower household income has been identified as an independent risk factor for the development of persistent asthma among children [8] and for worse asthma outcomes [9]. Suboptimal asthma self-management and worse asthma outcomes have been associated with lower health literacy [10–12]. Air quality has also been linked to asthma severity [13,14], although the literature on its association with asthma incidence and prevalence is inconclusive [15,16].

The UK has a high asthma prevalence and burden [1] and persistent socioeconomic inequalities where the more deprived people had worse asthma outcomes including higher risk of emergency admissions and deaths due to asthma [3,17]. In Wales, previous studies found that severe asthma admissions were more likely in the most deprived areas [18,19], which was attributed to active or passive smoking, variations in disease management, and air pollution. However, asthma is mostly managed in primary care in the UK with hospital care reserved for those with more severe disease or, most commonly, in the context of asthma attacks. Therefore, in order to comprehensively investigate inequalities in asthma care in this country, there is a need to study asthma care provision across care sectors. There is currently sparse evidence on the socioeconomic inequalities in asthma-related primary care, prescribing, accident and emergency (A&E) use, and deaths across the different demographic groups.

In this study, we investigated socioeconomic variations in asthma care and outcomes in Wales, including asthma-related primary and secondary health service utilisation, prescribing, and mortality, using a multifaceted measure of socioeconomic deprivation as well as individual domains of deprivation.

## Methods

### Ethics and permission

Research ethics approval was not required as we only used anonymised data. The Secure Anonymised Information Linkage (SAIL) Databank independent Information Governance Review Panel approved the study as part of the Wales Asthma Observatory project.

### Study design and data sources

We undertook a national, linked primary and secondary care retrospective cohort study of people with asthma in Wales. The follow-up period was 5 years from January 1, 2013 to December 31, 2017.

We used anonymised linked person-level datasets about primary and secondary care and causes of death from the Wales-wide SAIL Databank [20,21]. SAIL has 100% coverage in Wales for secondary care and causes of death and receives data from over 76% of the Welsh general practices. S1 Text provides more details about data sources and study design, and S2 Text details criteria of patient selection. S1 Table lists the code sets used in the patient selection and extraction of variables.

## Variables

**Socioeconomic status.** We measured socioeconomic status using the 2011 version of the Welsh Index of Multiple Deprivation (WIMD), the official area-based measure of relative socioeconomic deprivation in Wales [22]. The WIMD 2011 was constructed from a weighted sum of 8 deprivation domains: income (23.5%), employment (23.5%), health (14.0%), education (14.0%), geographical access to services (10.0%), housing (5.0%), physical environment (5.0%), and community safety (5.0%). The WIMD 2011 is based on Lower Layer Super Output Areas (LSOAs) which is a small area geography designed by the UK Office for National Statistics (ONS) for census-related purposes with consistent population sizes (1,500 people on average) [22]. We linked the patients to the WIMD through their residential addresses (LSOAs of the 2001 Census [23]) during the follow-up period. Where more than 1 address existed, we selected the address with the longest duration within that period. Quintiles of score were coded from 1 for the most deprived to 5 for the least deprived. S1 Fig shows the distribution of the WIMD 2011 score and its quintiles.

**Asthma-related health service utilisation.** Asthma-related health service utilisation was measured as counts of the corresponding primary and secondary care events, length of stay (LOS) in hospital, and controller-to-total asthma medication ratio (AMR) during the follow-up period.

We defined an "asthma-related general practitioner (GP) consultation" as 1 or more Read codes that indicated asthma-related contact with primary care professionals.

An asthma review was defined as scheduled consultations to a primary care practice in which disease control is assessed, and management plan, prescriptions, and asthma self-management advice were reviewed. The British Thoracic Society/the Scottish Intercollegiate Guidelines Network (BTS/SIGN) Guidelines on the Management of Asthma recommends that asthma reviews be arranged at least annually [24]. We identified routine asthma reviews from the Welsh Longitudinal General Practice (WLGP) dataset using codes of annual review, medication review, follow-up, monitoring by nurse, and review using the Royal College of Physicians' 3 Questions for Asthma [25].

We identified asthma-related A&E attendances with primary or secondary diagnosis of asthma from the Emergency Department Data Set using the dataset-specific asthma code 14A.

We identified asthma admissions from the Patient Episode Database for Wales as those with a primary diagnosis of asthma (J45) or status asthmaticus (J46) coded using the 10th Revision of the International Classification of Diseases (ICD-10). Among these, emergency admissions were defined as coming via A&E departments, urgent referrals from GPs, consultant clinics, bed bureaus, or NHS Direct [26].

**Asthma medication ratio.** The AMR, the ratio of controller-to-total asthma prescriptions, has been developed in the US as a surrogate quality measure of guideline adherence and is associated with patient outcomes and health service utilisation [27]. AMR calculation included counts of inhaled corticosteroids (ICS), ICS-long-acting beta adrenoceptor agonist (LABA) combination inhalers, sodium cromoglicate, and nedocromil as controller prescriptions, and short-acting beta agonist (SABA) inhalers as rescue prescriptions over the follow-up period. The formula was (ICS + ICS_LABA + sodium cromoglicate + nedocromil)/(ICS + ICS_LABA + sodium cromoglicate + nedocromil + SABA).

**Asthma-related death.** We used 2 definitions for asthma deaths: (1) deaths with any mention of asthma (an ICD-10 code of J45 or J46) in the death record; and (2) deaths with asthma as the underlying cause. Unlike the other outcomes, we analysed asthma deaths in a wider cohort of people with asthma diagnosed before the study period (see S2 Text).

## Statistical analysis

For the source population, we described demographics, point prevalence of ever being diagnosed with asthma (i.e., having an asthma diagnosis Read code) on January 1, 2013, and the period prevalence of ever-diagnosed currently treated asthma during 2013 (having an asthma diagnosis Read code ever and at least 1 asthma prescription code during 2013).

For the study cohort, we calculated the distribution, by the WIMD quintile, of age, gender, receipt of asthma prescription categories, and the health service utilisation variables. For AMR, we excluded patients who did not receive any of the prescriptions in the formula.

For each health service utilisation count variable, we fitted a negative binomial (NB) generalised linear model using the glm.nb function from the MASS package (version 7.3–51.4) [28]. We considered the least deprived quintile (WIMD 5) as the reference group. We adjusted the models for gender and age at the start of follow-up. We treated LOS in hospital as a count variable (count of days) for which the model coefficient represents the incidence rate ratio (IRR) of incurring an additional day in hospital in a given quintile compared with the least deprived quintile. Model fit was examined using quantile–quantile (Q–Q) plots of raw residuals and rootograms [29]. These models were specified a priori. However, we performed 2 separate sensitivity analyses; in the first one, we removed the condition of having continuous asthma treatment over the follow-up period, and in the second one, we removed the condition of continuous follow-up in the primary care dataset. We also modelled these count outcome variables using generalised additive models (GAMs) using the R package mgcv (version 1.8–31) [30]. We initially estimated global smooths for both the overall WIMD score and age. Then, we explored the within-gender effect of the overall WIMD score by estimating separate smooths for males and females. From these, we calculated a difference smooth [31] to explore the between-gender variation in the effect of the overall WIMD score. We then modelled the interaction between the overall WIMD score and age using a full tensor product smooth (mgcv::te). The count variables were also modelled separately against the score of each WIMD domains, controlled for age and gender.

We compared the mean AMR between the most and least deprived quintiles using Welch $t$ test. Then, we fitted a GAM using the beta regression family (mgcv::betar), adjusted for age and gender, to explore the associations with the overall WIMD score.

We modelled each definition of asthma deaths using logistic regression of WIMD quintile adjusted for gender and age. The effect of gender was then examined in separate models within each WIMD quintile. We then fitted binomial GAM models, including global smooths for the overall WIMD score and age, smooths by gender, and a difference smooth by gender, and compared the overall fitted risk of asthma death between males and females across the overall WIMD score. Finally, asthma deaths were then modelled separately against the score of each WIMD domains, controlled for age and gender.

All the GAMs in this paper used a thin plate regression spline as a smoothing basis and the fast restricted maximum likelihood (fREML) computation as a smoothing parameter estimation method.

We compared the ratio of overall emergency-to-total hospitalisations for asthma between the WIMD quintiles using equality of proportions test.

We used a confidence level of 95% ($p < 0.05$, 2-sided) throughout the study. All data analysis was performed in R 4.0.2.

## Reporting and supporting reproduction

This study is reported as per the Strengthening the Reporting of Observational Studies in Epidemiology (STROBE) [32] and the REporting of studies Conducted using Observational Routinely-collected health Data (RECORD) [33] guidelines (S1 Checklist).

The computer code used for data extraction and analysis is publicly available on GitHub [34].

## Study planning

Investigating asthma inequalities in Wales was a planned part of the first author's doctoral thesis about creating and utilising the Wales Asthma Observatory. There was no prospective protocol for the study. Health service utilisation count regression models against the WIMD 2011 score quintile were specified a priori, and preliminary results for an earlier follow-up period were previously published [35,36]. To provide a more comprehensive picture about inequalities, the analysis was developed to include wider outcomes such as asthma emergency admissions, prescribing, and deaths as well as the WIMD 2011's individual domains as predictors. The GAMs were developed in response to a peer review request to investigate nonlinear associations.

## Results

Table 1 shows characteristics of the source population (*n* = 2,871,257). The prevalence of ever-diagnosed asthma in the source population at the beginning of 2013 was 11.9% (95% CI, 11.8 to 11.9), with 12.5% (12.5 to 12.6) in the most deprived areas and 11.7% (11.6 to 11.8) in the least deprived areas. The prevalence of ever-diagnosed currently treated asthma during 2013 was 7.2% (7.1 to 7.2), ranging from 7.7% (7.6 to 7.8) to 6.8% (6.7 to 6.9) in the most and least deprived areas, respectively.

The study cohort included 106,926 patients (534,630 person-years) with ever-diagnosed, continuously treated asthma. Patient selection flowchart is shown in S2 Text, and Table 2 shows the patients' characteristics. Females comprised 56.3% of patients (58.9% to 54.2% in the most and least deprived areas). Mean age was 47.5 years (SD = 20.3).

Patients in the most deprived quintile (WIMD 1) represented the highest proportion (23.4%) of the study cohort and were younger overall (mean age = 45.6 years, SD = 20.1) than those in the least deprived areas (48.8 years, SD = 20.4). At least 1 asthma-related GP consultation and 1 asthma review was recorded over the follow-up period for 98.5% and 95.0% of patients, respectively. Only 2.4% and 3.6% had asthma-related A&E attendances and hospitalisations during the follow-up period.

Table 3 and Fig 1 show the estimated associations of the WIMD 2011 quintile with asthma-related primary and secondary care utilisation controlled for age and gender. The count regression models showed good fit to the data (see S2 Fig). The estimates did not significantly change by relaxing the patient selection criteria to also include patients with any asthma treatment status or any follow-up periods in the primary care dataset (see S3 Text).

**Table 1. Characteristics of the source population in 2013 across the WIMD 2011 quintiles.**

| WIMD quintile | Denominator | Ever-diagnosed asthma | | Ever-diagnosed currently treated asthma | |
|---|---|---|---|---|---|
| | | Cases | Prevalence % (95% CI) | Cases | Prevalence % (95% CI) |
| Most deprived | 481,660 | 60,441 | 12.5 (12.5, 12.6) | 37,142 | 7.7 (7.6, 7.8) |
| Next most deprived | 442,844 | 53,531 | 12.1 (12.0, 12.2) | 32,719 | 7.4 (7.3, 7.5) |
| Middle deprivation | 457,244 | 52,917 | 11.6 (11.5, 11.7) | 32,307 | 7.1 (7.0, 7.1) |
| Next least deprived | 399,188 | 45,763 | 11.5 (11.4, 11.6) | 27,344 | 6.8 (6.8, 6.9) |
| Least deprived | 467,111 | 54,649 | 11.7 (11.6, 11.8) | 31,865 | 6.8 (6.7, 6.9) |
| All | 2,248,047 | 267,301 | 11.9 (11.8, 11.9) | 161,377 | 7.2 (7.1, 7.2) |

CI, confidence interval; WIMD, Welsh Index of Multiple Deprivation.

**Table 2. Characteristics of the study cohort.**

| | Most deprived (WIMD 1) | Next most deprived (WIMD 2) | Middle deprivation (WIMD 3) | Next least deprived (WIMD 4) | Least deprived (WIMD 5) | All |
|---|---|---|---|---|---|---|
| Number of patients | 24,999 (23.4%) | 21,921 (20.5%) | 21,731 (20.3%) | 17,716 (16.6%) | 20,559 (19.2%) | 106,926 (100.0%) |
| Females | 58.9% | 57.0% | 55.5% | 55.2% | 54.2% | 56.3% |
| Age | 45.6 (20.1) | 46.9 (20.1) | 48.0 (20.5) | 48.6 (20.4) | 48.8 (20.4) | 47.5 (20.3) |
| Asthma-related GP consultations | 1.33, 1.2 (0.8–1.8) | 1.35, 1.2 (0.8–1.8) | 1.35, 1.2 (0.8–1.8) | 1.36, 1.2 (0.8–1.8) | 1.35, 1.2 (0.8–1.8) | 1.35, 1.2 (0.8–1.8) |
| | 98.3% | 98.5% | 98.5% | 98.6% | 98.9% | 98.5% |
| Asthma reviews | 0.77, 0.8 (0.4–1) | 0.76, 0.8 (0.4–1) | 0.75, 0.8 (0.4–1) | 0.74, 0.8 (0.4–1) | 0.79, 0.8 (0.6–1) | 0.76, 0.8 (0.4–1) |
| | 94.2% | 94.7% | 94.7% | 95.1% | 96.4% | 95.0% |
| Asthma-related A&E attendances | 0.008, 0 (0–0) | 0.007, 0 (0–0) | 0.007, 0 (0–0) | 0.006, 0 (0–0) | 0.005, 0 (0–0) | 0.007, 0 (0–0) |
| | 2.5% | 2.5% | 2.6% | 2.3% | 2.0% | 2.4% |
| Asthma hospitalisations | | | | | | |
| Total | 0.021, 0 (0–0) | 0.015, 0 (0–0) | 0.014, 0 (0–0) | 0.010, 0 (0–0) | 0.009, 0 (0–0) | 0.014, 0 (0–0) |
| | 4.4% | 3.9% | 3.4% | 3.0% | 2.8% | 3.6% |
| Emergency | 0.014, 0 (0–0) | 0.012, 0 (0–0) | 0.010, 0 (0–0) | 0.009, 0 (0–0) | 0.008, 0 (0–0) | 0.011, 0 (0–0) |
| | 4.2% | 3.7% | 3.3% | 3.0% | 2.7% | 3.4% |
| LOS | 0.055, 0 (0–0) | 0.052, 0 (0–0) | 0.041, 0 (0–0) | 0.035, 0 (0–0) | 0.032, 0 (0–0) | 0.044, 0 (0–0) |
| Prescriptions | | | | | | |
| SABA inhalers | 7.26, 6.6 (3.4–10.6) | 6.45, 5.6 (2.8–9.6) | 5.74, 4.6 (2.2–8.4) | 5.27, 4.0 (2–7.6) | 4.63, 3.4 (1.8–6.4) | 5.95, 4.8 (2.4–8.8) |
| | 98.5% | 97.9% | 97.3% | 97.4% | 97.5% | 97.7% |
| ICS inhalers | 2.15, 0.0 (0–3.4) | 2.03, 0.0 (0–3.2) | 1.94, 0.0 (0–3.0) | 1.92, 0.2 (0–3.0) | 1.84, 0.2 (0–2.8) | 1.99, 0.0 (0–3.0) |
| | 48.5% | 49.0% | 49.9% | 51.1% | 51.1% | 49.8% |
| ICS-LABA inhalers | 5.12, 4.4 (0–9.4) | 4.78, 4.0 (0–8.6) | 4.47, 3.4 (0–8.0) | 4.12, 3.0 (0–7.2) | 3.86, 2.8 (0–6.8) | 4.51, 3.4 (0–8.2) |
| | 68.4% | 66.9% | 65.1% | 63.4% | 63.1% | 65.6% |
| Sodium cromoglicate | 0.0, 0 (0–0) | 0.0, 0 (0–0) | 0.0, 0 (0–0) | 0.0, 0 (0–0) | 0.0, 0 (0–0) | 0.0, 0 (0–0) |
| | 0.0% | 0.0% | 0.1% | 0.1% | 0.1% | 0.1% |
| Nedocromil | 0.0, 0 (0–0) | 0.0, 0 (0–0) | 0.0, 0 (0–0) | 0.0, 0 (0–0) | 0.0, 0 (0–0) | 0.0, 0 (0–0) |
| | 0.0% | 0.1% | 0.0% | 0.0% | 0.0% | 0.0% |
| LTRA | 1.09, 0 (0–0) | 1.01, 0 (0–0) | 0.86, 0 (0–0) | 0.76, 0 (0–0) | 0.71, 0 (0–0) | 0.90, 0 (0–0) |
| | 17.8% | 16.5% | 15.5% | 14.2% | 14.5% | 15.8% |
| Theophylline | 0.27, 0 (0–0) | 0.24, 0 (0–0) | 0.20, 0 (0–0) | 0.18, 0 (0–0) | 0.10, 0 (0–0) | 0.20, 0 (0–0) |
| | 3.0% | 2.7% | 2.4% | 2.2% | 1.3% | 2.4% |
| Oral corticosteroids | 0.73, 0.2 (0–0.6) | 0.68, 0.2 (0–0.6) | 0.70, 0.2 (0–0.6) | 0.66, 0.2 (0–0.6) | 0.52, 0 (0–0.4) | 0.66, 0.2 (0–0.6) |
| | 54.6% | 53.1% | 52.2% | 51.5% | 47.0% | 51.8% |
| AMR | 0.50, 0.50 (0.44–0.56) | 0.52, 0.50 (0.44–0.61) | 0.53, 0.51 (0.44–0.66) | 0.54, 0.51 (0.44–0.67) | 0.56, 0.53 (0.45–0.71) | 0.53, 0.51 (0.44–0.64) |

Age at the start of follow-up is presented in years as mean (SD). Average health resource utilisation and prescriptions per year over the 5-year follow-up period is presented as mean, median (Q1–Q3), and percentage of patients with at least 1 event over the 5-year follow-up period. AMR is the ratio of controller asthma medications to total asthma medications [27].

A&E, accident and emergency; AMR, asthma medication ratio; GP, general practitioner; ICS, inhaled corticosteroid; IQR, interquartile range; LABA, long-acting beta adrenoceptor agonist; LOS, length of stay; LTRA, leukotriene receptor antagonists; SABA, short-acting beta agonist; SD, standard deviation; WIMD, Welsh Index of Multiple Deprivation.

## Asthma-related primary care consultations and reviews

The most deprived quintile (WIMD 1) had %1.9 fewer GP consultations (IRR = 0.98 [95% CI, 0.97 to 0.99], $p$-value < 0.001) and %2.0 fewer routine asthma reviews (IRR = 0.98 [95% CI, 0.97 to 0.99], $p$-value < 0.001) per patient compared with the least deprived quintile (WIMD

**Table 3. IRRs with 95% CIs of asthma health service utilisation in each of the WIMD 2011 quintiles relative to the least deprived quintile (fifth quintile), controlled for age and gender.**

| | Asthma GP consultations | | Asthma reviews | | Asthma A&E attendances | | Asthma admissions | | | | | |
| | | | | | | | Total | | Emergency | | LOS | |
| | IRR (95% CI) | *p*-value | IRR (95% CI) | *p*-value | IRR (95% CI) | *p*-value | IRR (95% CI) | *p*-value | IRR (95% CI) | *p*-value | IRR (95% CI) | *p*-value |
|---|---|---|---|---|---|---|---|---|---|---|---|---|
| 1. Most deprived | 0.981 (0.972, 0.991) | <0.001 | 0.980 (0.970, 0.990) | <0.001 | 1.269 (1.103, 1.459) | 0.001 | 1.977 (1.740, 2.246) | <0.001 | 1.559 (1.385, 1.756) | <0.001 | 1.640 (1.385, 1.942) | <0.001 |
| 2. Next most deprived | 0.996 (0.986, 1.006) | 0.448 | 0.964 (0.953, 0.974) | <0.001 | 1.229 (1.063, 1.421) | 0.005 | 1.450 (1.268, 1.658) | <0.001 | 1.407 (1.244, 1.592) | <0.001 | 1.568 (1.318, 1.867) | <0.001 |
| 3. Middle deprivation | 1.003 (0.993, 1.013) | 0.508 | 0.954 (0.943, 0.964) | <0.001 | 1.244 (1.075, 1.440) | 0.003 | 1.478 (1.292, 1.691) | <0.001 | 1.196 (1.054, 1.357) | 0.006 | 1.293 (1.085, 1.540) | 0.004 |
| 4. Next least deprived | 1.008 (0.998, 1.019) | 0.130 | 0.942 (0.931, 0.953) | <0.001 | 1.183 (1.013, 1.381) | 0.034 | 1.162 (1.006, 1.343) | 0.041 | 1.095 (0.957, 1.253) | 0.185 | 1.094 (0.909, 1.316) | 0.341 |
| Age (years) | 0.999 (0.999, 0.999) | <0.001 | 1.003 (1.003, 1.003) | <0.001 | 0.965 (0.962, 0.967) | <0.001 | 0.979 (0.977, 0.981) | <0.001 | 0.979 (0.977, 0.981) | <0.001 | 0.994 (0.991, 0.996) | <0.001 |
| Gender (female) | 1.024 (1.017, 1.031) | <0.001 | 1.050 (1.043, 1.058) | <0.001 | 1.624 (1.481, 1.781) | <0.001 | 1.606 (1.477, 1.747) | <0.001 | 1.667 (1.541, 1.804) | <0.001 | 1.982 (1.773, 2.216) | <0.001 |
| Intercept | 6.938 (6.864, 7.012) | <0.001 | 3.316 (3.277, 3.355) | <0.001 | 0.091 (0.079, 0.105) | <0.001 | 0.092 (0.080, 0.106) | <0.001 | 0.075 (0.066, 0.086) | <0.001 | 0.140 (0.116, 0.169) | <0.001 |

A&E, accident and emergency; CI, confidence interval; GP, general practitioner; IRR, incidence rate ratio; LOS, length of hospital stay; WIMD, Welsh Index of Multiple Deprivation.

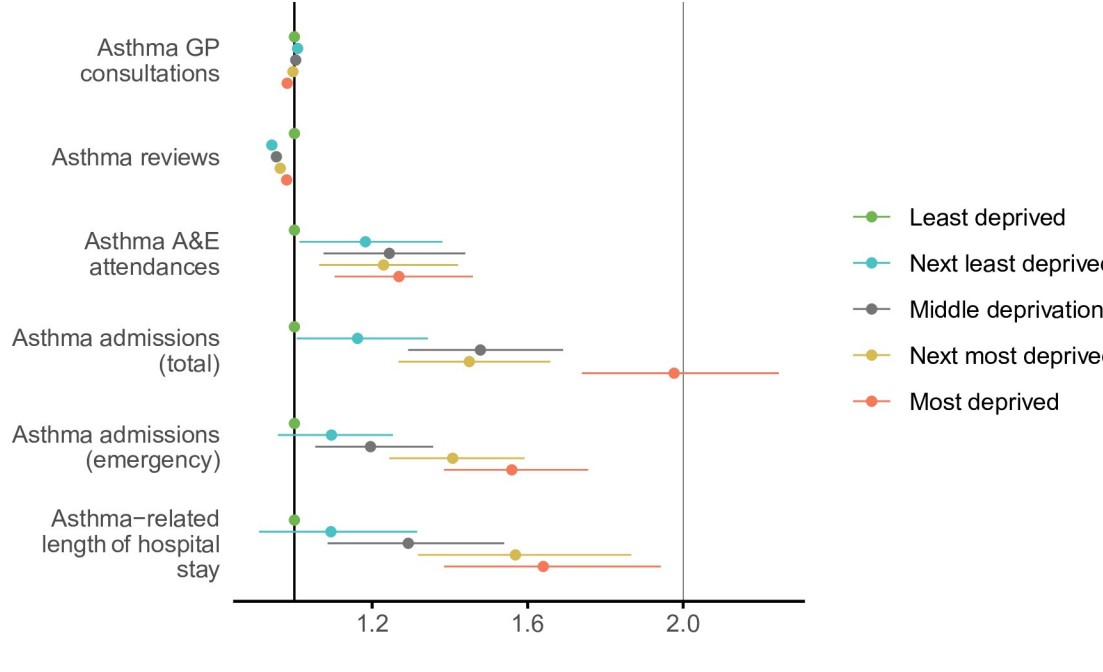

**Fig 1. IRRs with 95% CIs of asthma health service utilisation in each of the WIMD 2011 quintiles relative to the least deprived quintile, controlled for age and gender.** Note: The CIs for GP consultations and review are extremely narrow and may not be obvious in the plot. A&E, accident and emergency; CI, confidence interval; GP, general practitioner; IRR, incidence rate ratio; WIMD, Welsh Index of Multiple Deprivation.

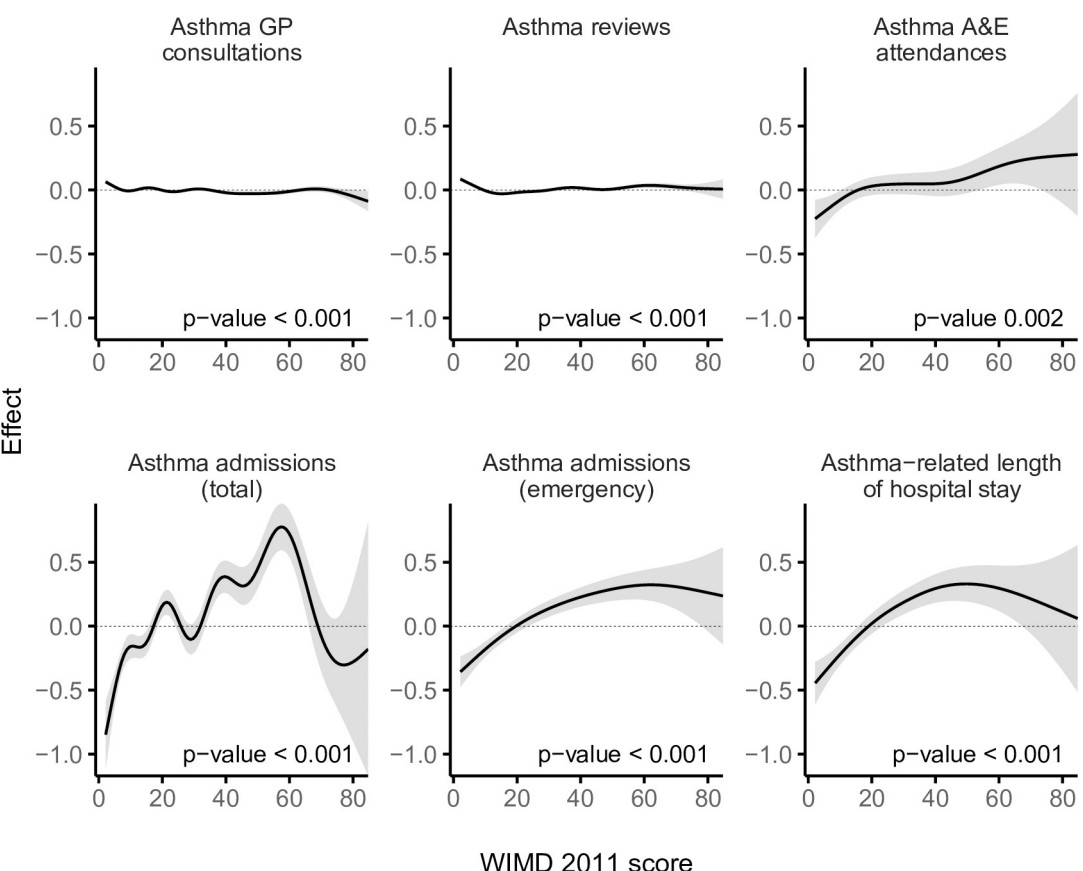

**Fig 2. Smooths from GAMs showing the association of the WIMD 2011 score with asthma-related health service utilisation.** The shaded area represents the 95% CI of effect. A&E, accident and emergency; CI, confidence interval; GAM, generalised additive model; GP, general practitioner; WIMD, Welsh Index of Multiple Deprivation.

5). The WIMD quintiles 2 to 4 were also associated with slightly fewer asthma reviews than the least deprived quintile. The corresponding GAMs support these association patterns (Fig 2). Less geographical access to services was associated with slightly more asthma GP consultations and slightly fewer asthma reviews (Fig 3). Overall, females had 2.4% more asthma-related GP consultations (IRR = 1.02 [1.02 to 1.03], *p*-value < 0.001) and 5.0% more asthma reviews (1.05 [1.04 to 1.06], *p*-value < 0.001) than males. However, these gender gaps had minor variations across the overall WIMD score (Figs 4 and 5) and age: In middle age, females had slightly higher rates than males, whereas males had slightly higher rates among children and the older adults. In both genders, however, there was a minor variation across age, with asthma reviews being lowest in middle age, whereas the younger people had slightly more asthma-related GP consultations (Fig 6).

## Asthma prescriptions in primary care

Patients in the most deprived areas had more primary care asthma prescriptions, at 17.6 prescriptions per year, compared with 12.0 in the least deprived areas. A socioeconomic gradient existed for all the classes of asthma medications (Table 2). On average, people from the most deprived areas received 7.3 reliever (SABA) inhalers and 7.3 controller prescriptions of ICS, ICS-LABA, sodium cromoglicate, and/or nedocromil per year, whereas those in the least deprived areas received 4.6 relievers and 5.7 controllers. Patients in the most deprived areas

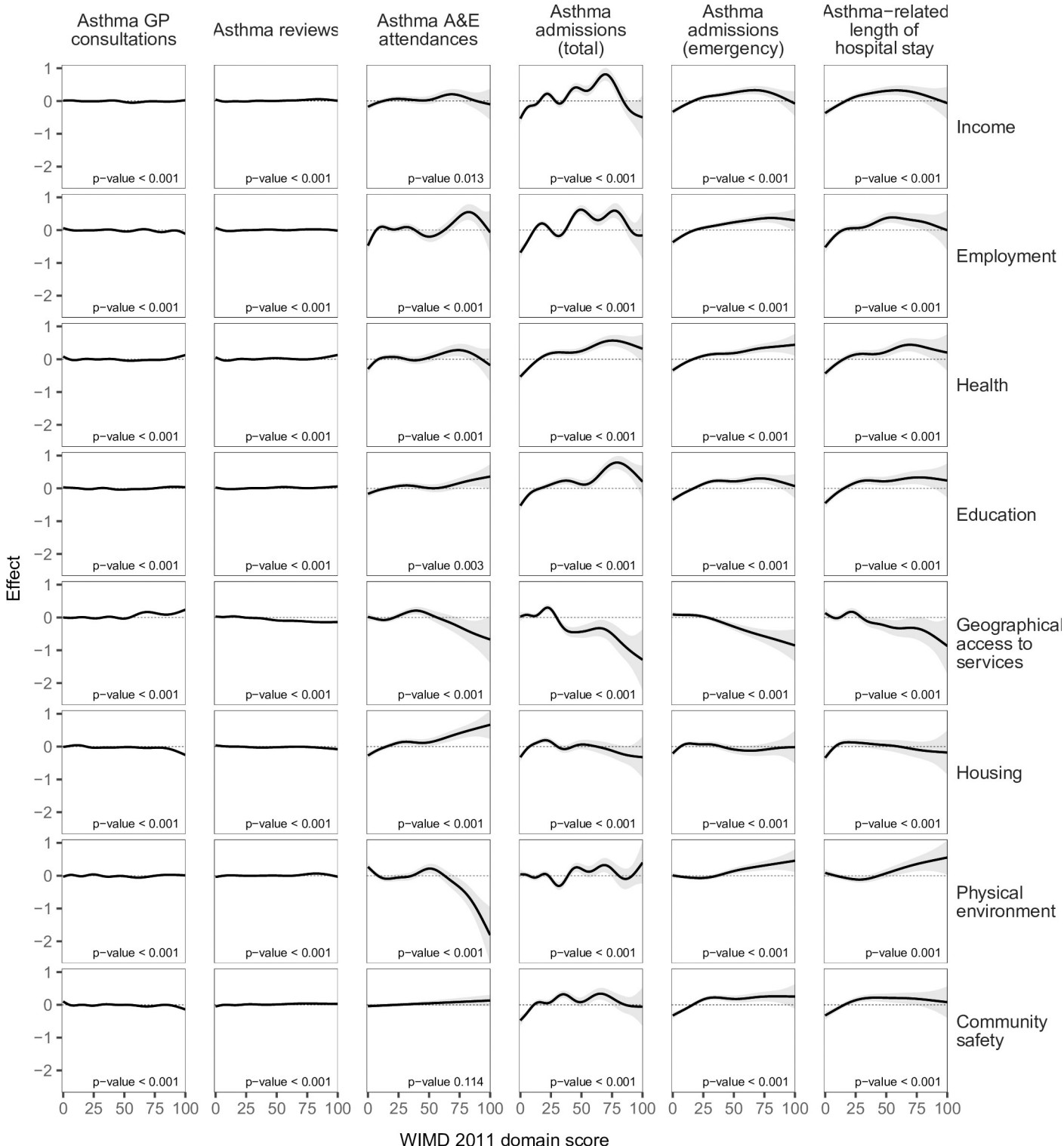

**Fig 3. Smooths from separate GAMs of asthma-related health service utilisation variables against scores of the WIMD 2011 domains, controlled for age and gender.** The shaded area represents the 95% CI of effect. A&E, accident and emergency; CI, confidence interval; GAM, generalised additive model; GP, general practitioner; WIMD, Welsh Index of Multiple Deprivation.

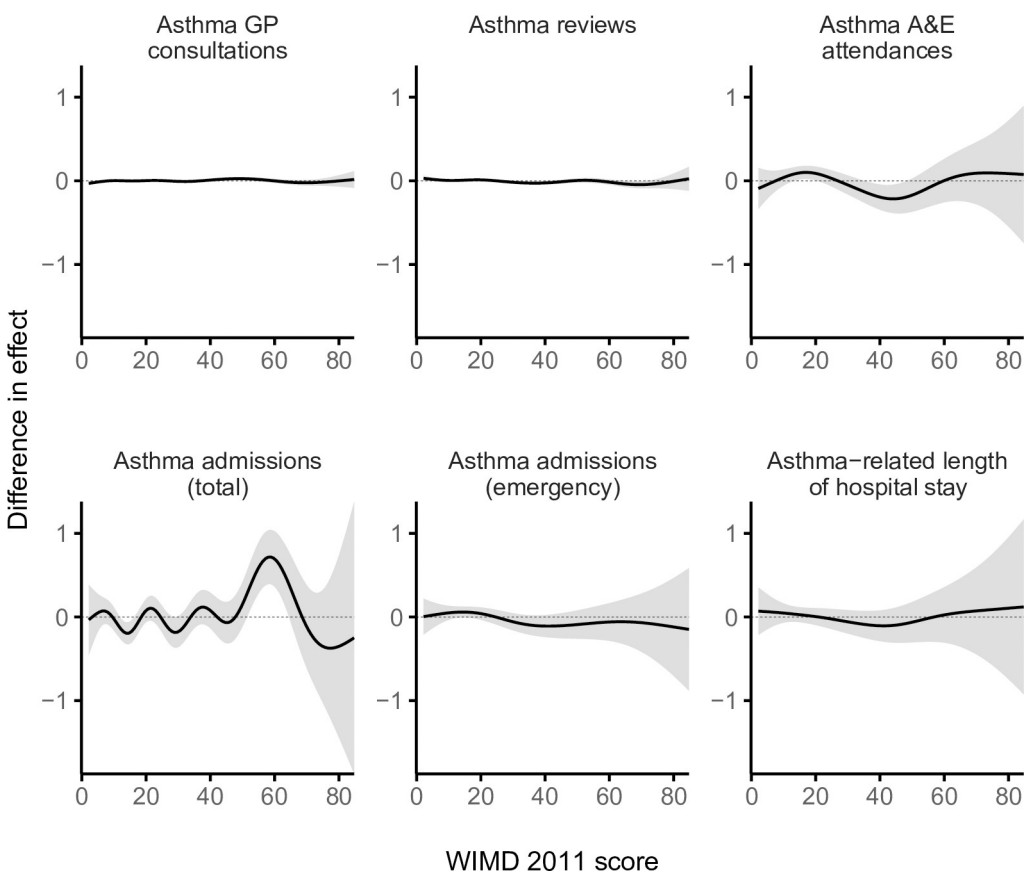

**Fig 4. Difference smooths from GAMs showing gender variations in the effect of the WIMD 2011 score on asthma-related health service utilisation.** The shaded area represents the 95% CI of effect. A positive difference means a higher effect in females than males. The gender gap was stable for asthma primary care consultations and reviews and variable for A&E attendances, emergency admissions, and LOS. A&E, accident and emergency; CI, confidence interval; GAM, generalised additive model; GP, general practitioner; LOS, length of stay; WIMD, Welsh Index of Multiple Deprivation.

were 3.0 times more likely to have 12 or more SABA inhaler per year (risk ratio = 3.0 [2.8, 3.2], *p*-value < 0.001).

Mean AMR (controller-to-total medication ratio) was lower in the most deprived quintile (0.50) than in the least deprived quintile (0.56; Welch *t* test *p*-value < 0.001; ratio of means = 90.3% [89.6% to 91.0%]; absolute difference in means = 0.054 [0.050 to 0.058]). The GAM showed that AMR generally decreased with higher WIMD score up to around the overall WIMD score of 40 before increasing slightly in the higher WIMD scores (Fig 7). However, AMR distribution below 0.5 was similar across the WIMD quintiles, whereas fewer patients had AMR >0.5 in the more deprived quintiles (see the empirical cumulative distribution function plot in S3 Fig). The variation of AMR across age was greater than across the WIMD score, with the late teens and early 20s having the lowest values. There was no statistically significant difference in AMR between males and females (odds ratio: 1.003 [0.989 to 1.017], *p*-value = 0.721).

## Asthma-related A&E attendances

The most deprived areas had 26.9% more A&E attendances than the least deprived areas (IRR = 1.27 [1.10 to 1.46], *p*-value = 0.001). The IRR decreased to 1.23 [1.07, 1.41] after

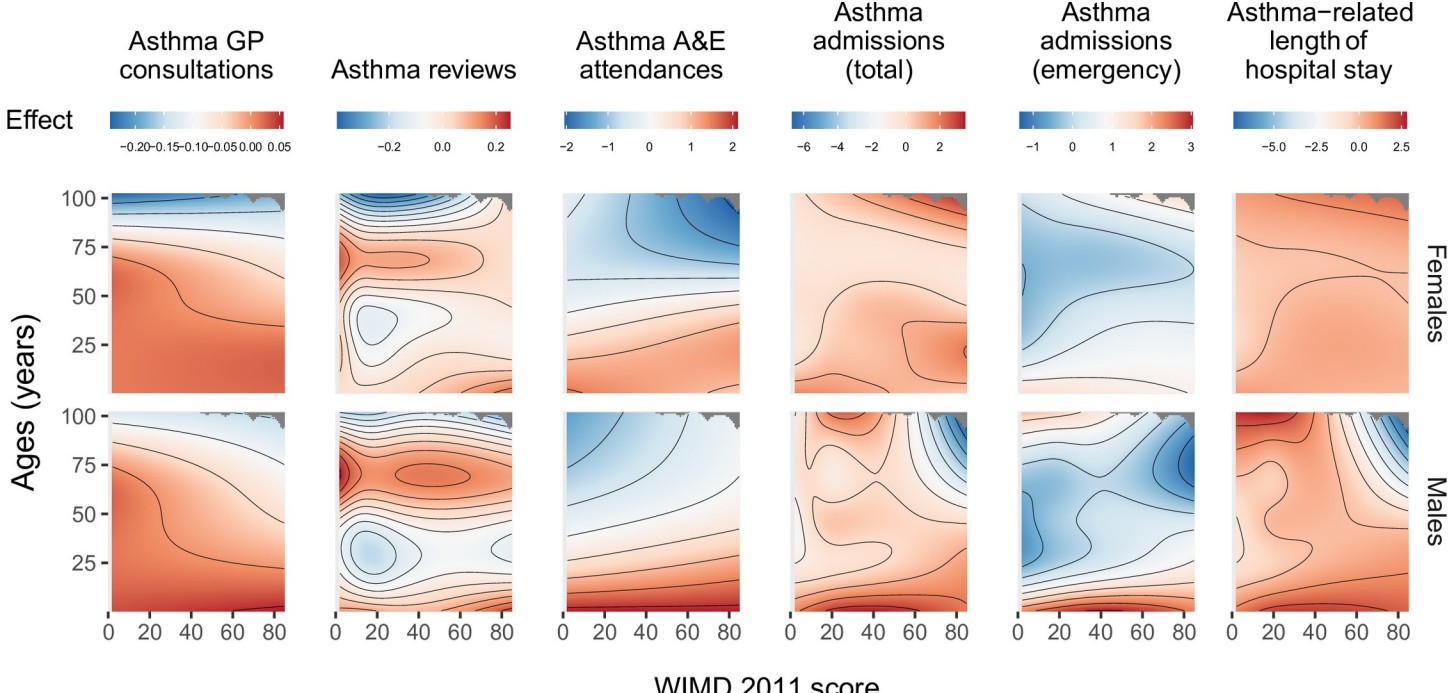

**Fig 5. Full tensor product smooths from GAMs of asthma-related health service utilisation showing interaction between the WIMD 2011 score and age in males and females.** The colour represents the partial effect (blue = lower, red = higher). The models did not include separate smooths for the WIMD 2011 score and age. A&E, accident and emergency; CI, confidence interval; GAM, generalised additive model; GP, general practitioner; WIMD, Welsh Index of Multiple Deprivation.

controlling for AMR (*p*-value = 0.004). No gradient existed across the deprivation quintiles, but there was a contrast between the least deprived quintile and the other more deprived quintiles together, a pattern also seen in the corresponding GAM (Fig 2). Lower education levels and worse housing were consistently associated with higher attendance rates. Higher rates were also seen towards lower levels of income, employment, and general health (Fig 3). However, rates were lower with less geographical access to services and worse physical environment.

Overall, females had 62.4% more asthma-related A&E attendances than males (IRR = 1.62 [1.48 to 1.78], *p*-value < 0.001). However, the gender gap was variable across age and deprivation; in middle age, females had higher rates than males, whereas males had higher rates in childhood and also among the older adults living in the most deprived areas (Figs 5 and 8). Overall, rates were highest in the youngest patients and steeply decreased in older ages (Figs 5 and 6).

## Asthma hospitalisations

A steep socioeconomic gradient existed for emergency and total asthma-related hospitalisations (Figs 1 and 2). Patients in the most deprived quintile were 55.9% more likely to require emergency admissions for asthma (IRR = 1.56 [1.38 to 1.76], *p*-value < 0.001) than those in the least deprived quintile. The IRR decreased to 1.52 [1.35 to 1.72] after controlling for AMR (*p*-value < 0.001). Patients in the most deprived quintile were 97.7% more likely to require any hospitalisation (whether emergency or elective) for asthma (IRR = 1.98 [1.74 to 2.25], *p*-value < 0.001) than those in the least deprived quintile. Asthma patients in the most deprived areas had 64.0% longer asthma-related hospital stay than those in the least deprived areas

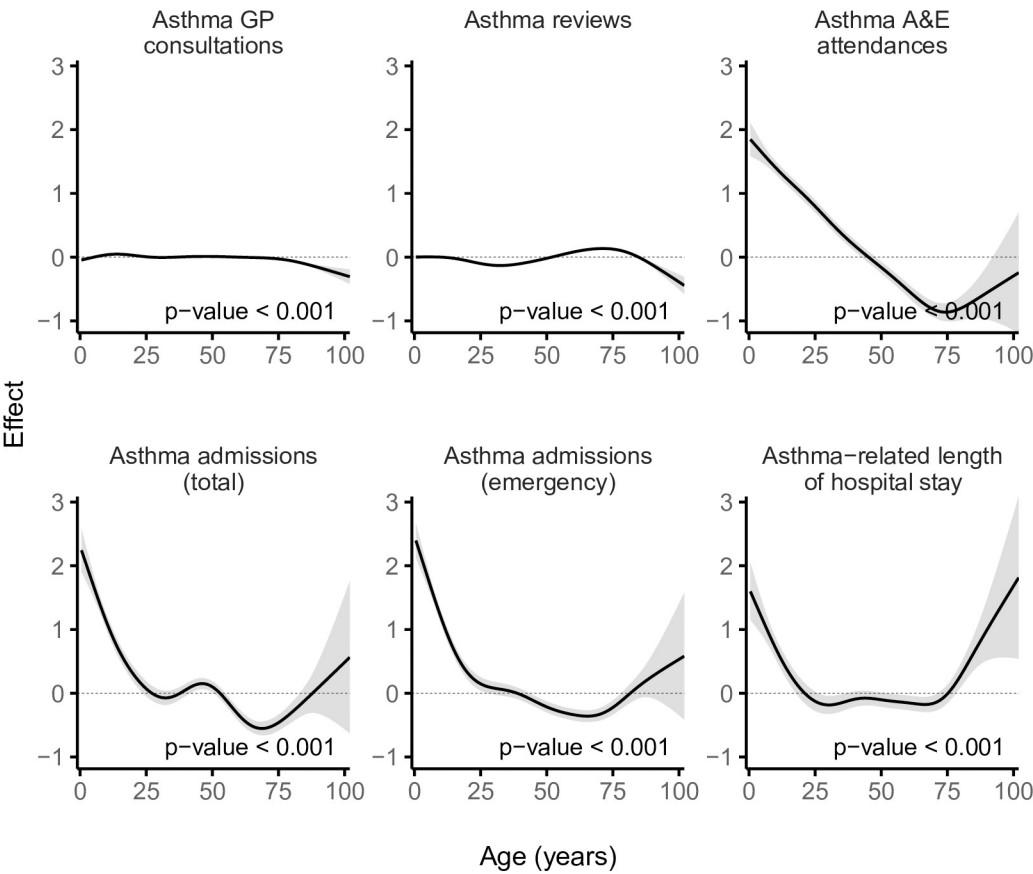

**Fig 6. Smooths from GAMs showing the variation in asthma-related health service utilisation across age.** The shaded area represents the 95% CI of effect. A&E, accident and emergency; CI, confidence interval; GAM, generalised additive model; GP, general practitioner.

(mean of 0.25 versus 0.15 days during the 5-year follow-up period, respectively; IRR = 1.64 [1.39 to 1.94], p-value < 0.001). The GAMs support these patterns, although they show variation within the most deprived quintile with the highest admission rates being around the

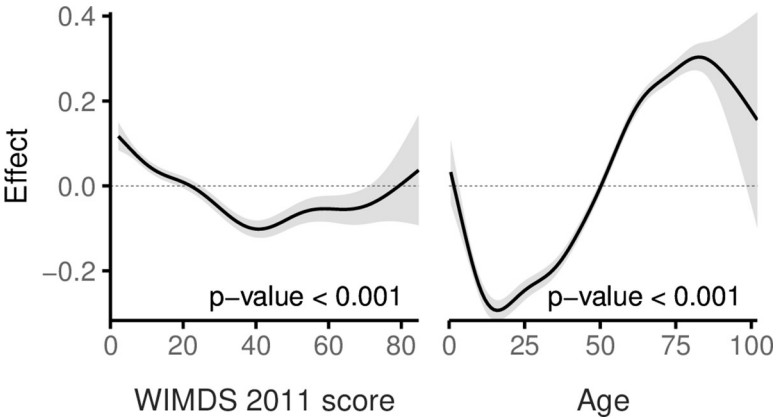

**Fig 7. A GAM of AMR by the WIMD 2011 score and age, adjusted for gender.** The shaded area represents the 95% CI of effect. AMR, asthma medication ratio; CI, confidence interval; GAM, generalised additive model; WIMD, Welsh Index of Multiple Deprivation.

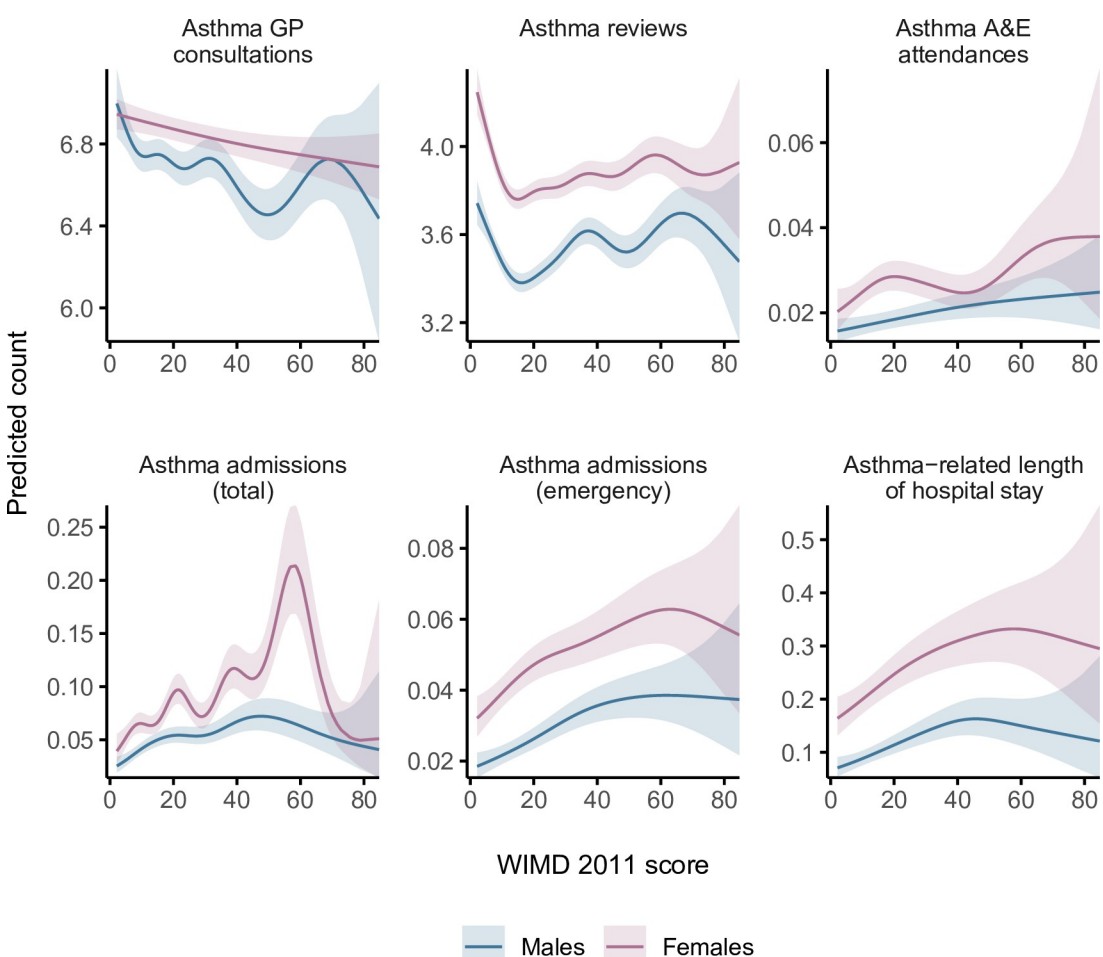

**Fig 8. Predicted 5-year counts of health service utilisation events for the median age in males and females based on GAMs of asthma-related health service utilisation by the WIMD 2011 score.** The shaded areas represent the 95% CIs of predicted counts. A&E = general practitioner; GP = general practitioner; WIMD, Welsh Index of Multiple Deprivation.

overall WIMD score of 60 (Fig 2). Higher rates of emergency admissions were associated with worse scores in all the WIMD domains except housing. However, the rates decreased with less geographical access to services (Fig 3). At the WIMD quintile level, however, the most deprived quintile had the lowest proportion of emergency-to-total asthma-related admissions (66.2%), whereas the least deprived quintile had the highest proportion (85.4%, S4 Fig). In the whole cohort, 57.1% of the nonemergency admissions were day cases with no overnight stay.

Overall, females had 66.7% more emergency admissions (1.67 [1.54 to 1.80], $p$-value < 0.001), 60.6% more total hospitalisations (1.61 [1.48 to 1.75], $p$-value < 0.001), and 98.2% longer hospital stay related to asthma (1.98 [1.77 to 2.22], $p$-value < 0.001) than males. However, the gender gap showed some variations across the overall WIMD score and age; females had higher admission rates and longer hospital stay in middle age and also among the most deprived older adults, whereas males had higher rates among children and also among the less deprived older adults (Figs 5 and 8).

Children, especially males, had higher asthma admissions rates and longer stay in hospital than the other age groups (Fig 6).

### Asthma-related deaths

In the wider cohort of 327,906 asthma patients, 543 had death with any mention of asthma and 207 had death with asthma as the underlying cause over the study period. Risk of death with either definitions generally increased with higher deprivation (Fig 9). Asthma patients in the most deprived quintile were 56.3% more likely to have death with any mention of asthma within 5 years than those in the least deprived areas (risk ratio, RR = 1.56 [1.18 to 2.07], $p$-value = 0.002, Table 4). When asthma deaths were identified by the underlying cause only, we could not detect differences between the least deprived and other quintiles, although deaths were associated with the overall WIMD score (Fig 9).

Among females, risk of asthma deaths generally increased with deprivation, whereas among males, the highest risk was in the middle WIMD quintile (around the score of 19, Fig 10). Females were generally at higher risk of asthma deaths (Fig 11). However, the gender gap varied across the overall WIMD score and was wider (i.e., females having higher risk) in the most deprived (risk ratios for deaths with asthma as the underlying cause, RR = 4.16 [1.76 to 9.80], $p$-value = 0.001) and in the least deprived quintiles (2.74 [1.20 to 6.24], $p$-value = 0.016) and diminished in the middle deprivation quintile where the WIMD score tended to have a stronger effect among males (Fig 12). However, among the younger patients in the least deprived areas, males had higher risk of death than females (Fig 13).

Generally, older age and higher deprivation were associated with high risk of asthma deaths. However, in males over the age of 75 years, lower deprivation was associated with higher risk (Fig 13).

Finally, deaths with asthma as the underlying cause were associated with lower income and employment levels, and there was weaker evidence that these deaths were associated with lower health and education levels. However, deaths with any mention of asthma were associated with lower scores in all those 4 domains (Fig 14).

## Discussion

### Main findings

We identified worse asthma outcomes for people in the most deprived areas of Wales across all stages of patient care. Compared with those in the least deprived areas, the most deprived

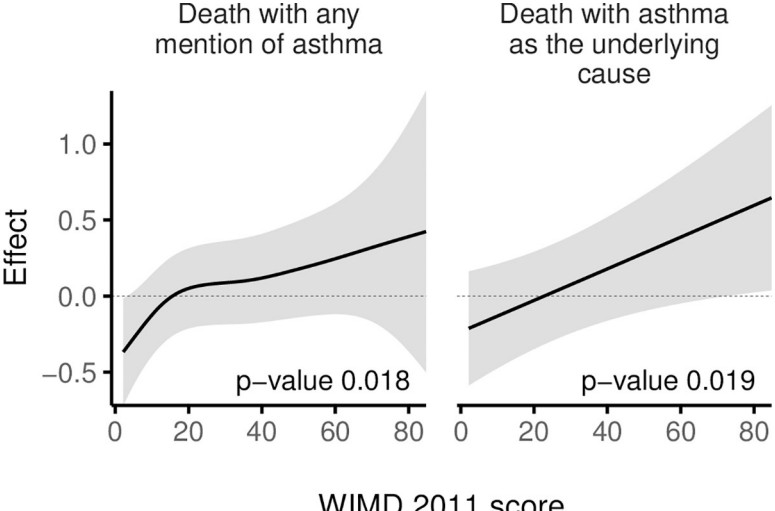

**Fig 9. Smooths from GAMs of asthma deaths by the WIMD 2011 score, controlled for age and gender.** The shaded area represents the 95% CI of effect. CI, confidence interval; GAM, generalised additive model; WIMD, Welsh Index of Multiple Deprivation.

**Table 4. Risk ratio of asthma deaths by age and gender within the WIMD 2011 score quintiles.**

| | Any mention of asthma | | Asthma as the underlying cause | |
|---|---|---|---|---|
| | Risk ratio (95% CI) | *p*-value | Risk ratio (95% CI) | *p*-value |
| **Deprivation quintile: the reference quintile is the least deprived (WIMD 5)** | | | | |
| Most deprived (WIMD 1) | 1.56 (1.18, 2.07) | 0.002 | 1.39 (0.91, 2.13) | 0.124 |
| Next most deprived (WIMD 2) | 1.45 (1.09, 1.92) | 0.010 | 1.17 (0.76, 1.82) | 0.470 |
| Middle deprivation (WIMD 3) | 1.36 (1.02, 1.80) | 0.034 | 0.97 (0.62, 1.52) | 0.895 |
| Next least deprived (WIMD 4) | 1.49 (1.13, 1.98) | 0.005 | 1.18 (0.76, 1.83) | 0.466 |
| Gender (female) | 1.21 (1.01, 1.45) | 0.042 | 1.52 (1.12, 2.07) | 0.007 |
| Age | 1.10 (1.09, 1.10) | <0.001 | 1.09 (1.08, 1.11) | <0.001 |
| **Within-quintile models** | | | | |
| *Most deprived* | | | | |
| Female | 1.84 (1.20, 2.81) | 0.005 | 4.16 (1.76, 9.80) | 0.001 |
| Age | 1.09 (1.07, 1.10) | <0.001 | 1.08 (1.06, 1.10) | <0.001 |
| *Next most deprived* | | | | |
| Female | 1.15 (0.77, 1.70) | 0.495 | 1.49 (0.76, 2.93) | 0.245 |
| Age | 1.09 (1.08, 1.11) | <0.001 | 1.08 (1.06, 1.11) | <0.001 |
| *Middle deprivation* | | | | |
| Female | 0.80 (0.55, 1.17) | 0.249 | 0.55 (0.28, 1.05) | 0.069 |
| Age | 1.09 (1.08, 1.10) | <0.001 | 1.10 (1.07, 1.12) | <0.001 |
| *Next least deprived* | | | | |
| Female | 1.07 (0.73, 1.59) | 0.721 | 1.12 (0.58, 2.14) | 0.741 |
| Age | 1.12 (1.10, 1.13) | <0.001 | 1.12 (1.09, 1.14) | <0.001 |
| *Least deprived* | | | | |
| Female | 1.58 (0.98, 2.54) | 0.059 | 2.74 (1.20, 6.24) | 0.016 |
| Age | 1.11 (1.09, 1.13) | <0.001 | 1.12 (1.09, 1.15) | <0.001 |

Age was measured in years.

CI, confidence interval; WIMD, Welsh Index of Multiple Deprivation.

group had slightly less asthma-related unscheduled primary care, slightly less structured proactive asthma care, and poorer quality of prescribing, but a markedly higher asthma-related A&E attendances, hospitalisations, and risk of asthma-related death. Clear socioeconomic gradients existed in emergency admissions, hospital days, and death. Moreover, higher deprivation was associated with more asthma prescriptions, higher risk of the excessive use of reliever inhalers, and prescribing of fewer controller medications relative to reliever inhalers. Lower levels of income, education, employment, and general health were generally associated with higher rates of asthma emergency care and death. However, patients in rural areas used less asthma emergency care. Lastly, females in middle age had higher asthma-related primary and secondary care utilisation than males and higher risk of deaths among all adults, although the inverse patterns were seen in children, and variable patterns existed in the older ages.

## Interpretation of findings and comparison with other studies

A complex relationship potentially exists between the WIMD 2011 and asthma outcomes. Asthma is among limiting, long-term illnesses that contribute to the WIMD 2011 health domain [22]. The WIMD also incorporates birth outcomes, including low birth weight and preterm delivery [22]—factors linked to maternal asthma severity and medications [37–39]. Asthma also potentially affects the WIMD education and employment domains as it is linked

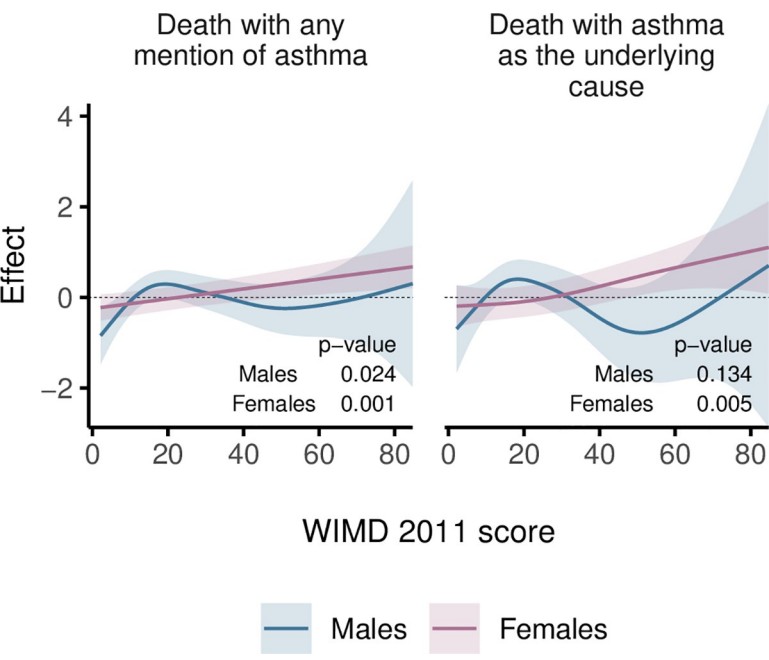

**Fig 10. Smooths from GAMs showing the association of the WIMD 2011 score on asthma death in males and females.** The shaded area represents the 95% CI of effect. CI, confidence interval; GAM, generalised additive model; WIMD, Welsh Index of Multiple Deprivation.

to school absenteeism [40] and job absenteeism and loss [41]. With the above direct and indirect links between asthma and the WIMD 2011, predicting asthma outcomes by the WIMD requires cautious interpretation of the findings. The socioeconomic variation in asthma-related primary care consultations and structured asthma review is marginal and may have no

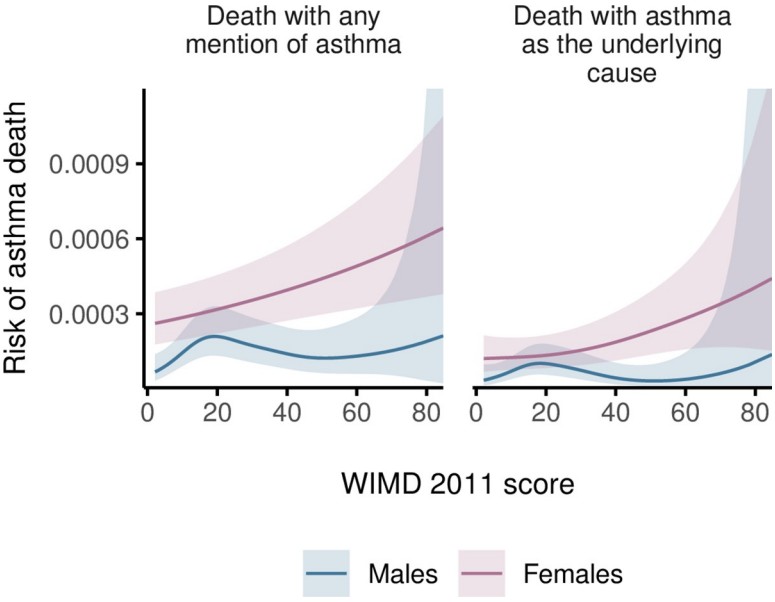

**Fig 11. Estimated 5-year risk of asthma deaths for median age in males and females by the WIMD 2011 score.** The shaded area represents the 95% CI of risk. CI, confidence interval; Welsh Index of Multiple Deprivation.

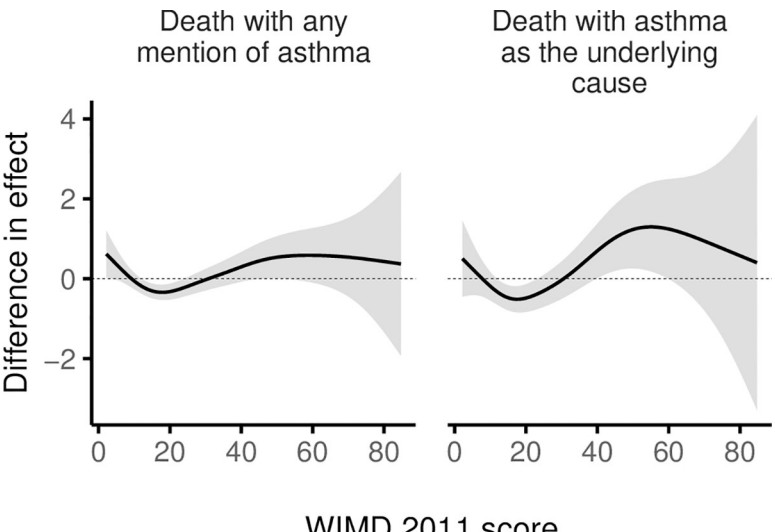

**Fig 12. Difference smooths from GAMs showing a varying gender gap in the effect of WIMD 2011 score on asthma deaths.** The shaded area represents the 95% CI of difference in effect. A positive difference means a higher effect in females than males. CI, confidence interval; GAM, generalised additive model; WIMD, Welsh Index of Multiple Deprivation.

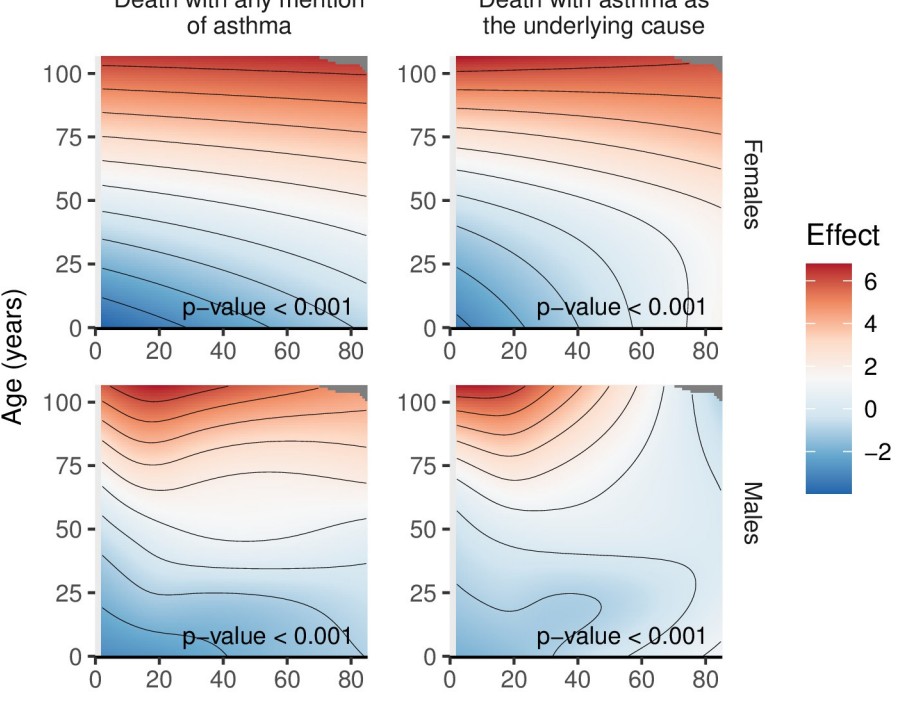

**Fig 13. Full tensor product smooths from GAMs of asthma deaths by the WIMD 2011 score and age in males and females.** The colour represents the partial effect (blue = lower, red = higher). The models did not include separate smooths for the WIMD 2011 score and age. Generally, higher deprivation and older age were associated with higher risk of asthma deaths. However, in males over the age of 75 years, lower deprivation was associated with higher risk. GAM, generalised additive model; WIMD, Welsh Index of Multiple Deprivation.

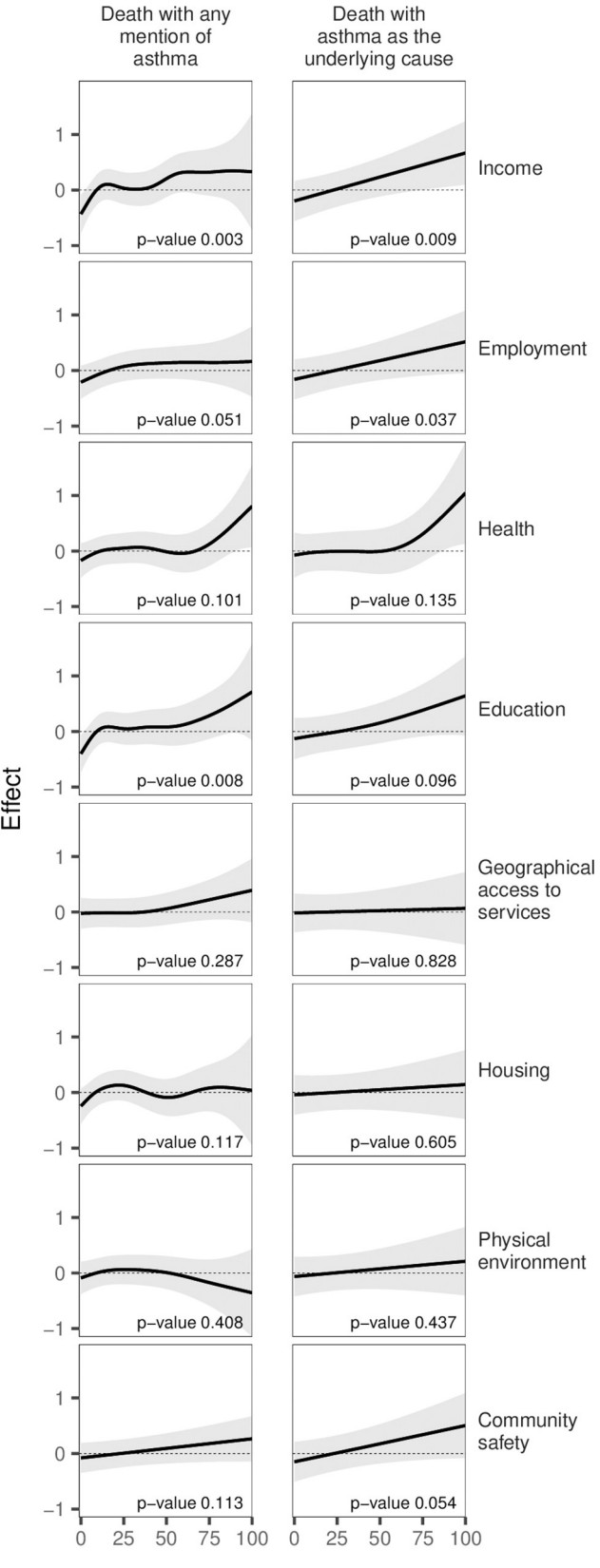

**Fig 14. GAMs of asthma death by the scores of WIMD 2011 domains, controlled for age and gender.** The shaded area represents the 95% CI of effect. CI, confidence interval; GAM, generalised additive model; WIMD, Welsh Index of Multiple Deprivation.

clinical significance. However, the wider gap in emergency asthma care and deaths indicates inequalities in disease severity and potentially in how the disease is managed. Higher asthma-related health service utilisation, especially in secondary care, often indicates more severe and/or uncontrolled disease [42].

The higher asthma-related A&E attendances in the most deprived areas might not be solely driven by more severe or worse controlled disease, but also by a complex host of factors including poorer inhaler technique and/or greater other medical/social problems (e.g., comorbidities) that worsens asthma experience. In addition, some of those A&E attendances could be driven by the tendency of people, particularly in the more deprived areas, to use A&E departments as primary care facilities. This could be possibly due to insufficient health literacy [43], which is consistent with the higher A&E attendances with lower education levels seen in our study, and/or due to GP practices and pharmacies being overcrowded or inaccessible [44–46]. Accordingly, the socioeconomic contrast in asthma-related A&E attendances could overestimate the gap in asthma severity and control. Nonetheless, this gap is still evident from the clear socioeconomic gradient in asthma-related emergency admissions and deaths. However, areas with fewer local services and transport nodes, i.e., rural areas, had lower A&E attendances and admissions for asthma and slightly higher unscheduled asthma GP consultations. This may suggest that emergency care could be less accessible by those who need it who instead seek it in primary care. However, it could also be explained by the inverse correlation between the geographical access to services and the income and education levels which, as our study shows, are, in turn, inversely associated with asthma emergency care.

The 1.56 [1.18 to 2.07] risk ratio of asthma-related death between the most deprived compared to the least deprived quintiles was an average across age groups. An analysis of asthma deaths in England found similar relative risk for patients ≥45 years old (IRRs between 1.30 and 1.37) [3]. However, in that analysis, this pattern was unexpectedly reversed among the younger patients (IRR = 0.81 [0.69, 0.96]), but this was not seen in our study. Instead, the interaction we modelled between the WIMD 2011 score and age showed that among the over 75-year-old males, higher deprivation was associated with lower risk of emergency asthma admissions and deaths.

The gap in asthma severity and control could be explained by factors related to the disease, patient, and healthcare. Lack of education opportunities may have resulted in lower educational attainment and health literacy. This may hinder asthma self-management, adherence to treatment, proper inhaler technique, and engagement in clinical decision-making [10–12], leading to poorer asthma control and higher dependency on healthcare, including a greater need for A&E attendances and emergency admissions [47]. The modestly higher use of both controller and reliever prescriptions with higher overall deprivation is consistent with a gradient in asthma severity and/or control. However, the corresponding inverse gradient of lower controller-to-total medication ratio—a measure of whether controllers are adequate relative to relievers—with higher deprivation suggests variations in prescribing, dosing, adherence, and/or asthma self-management [48].

Air pollution, a sub-domain in the WIMD 2011 [22], is a possible contributor to socioeconomic inequalities in asthma outcomes. A previous study in Wales found that independent measures of air pollution had weak to modest effects on "serious" asthma admissions—prolonged admissions or those followed by death from any cause [19]. While there is

contradictory literature about the effect of air pollution on asthma incidence and prevalence [15,16], it was associated with higher risk of exacerbations, especially in those who live or spend time close to busy roads [13,14].

Females, especially between the age of 16 and 60 years, are generally overrepresented in primary care [49]. Nonetheless, in our study, the gender gap in asthma-related primary care consultations was minimal. However, females had higher rates of asthma-related A&E attendances, admissions in middle age, and higher risk of asthma deaths among all adults, with the inverse pattern in childhood, which is generally consistent with other studies [50–54]. These patterns could be driven by gender differences in asthma development, disease experience, and outcomes [55,56].

### Strength and limitations

Our study has several strengths. We used objective, real-world, person-level data with high to complete nationwide representativeness to identify most people with asthma in Wales and individually measure their asthma-related health service use. Free-of-charge healthcare, including prescriptions, limits the potential bias of patients on low income avoiding healthcare access to minimise out-of-pocket expenses [57,58]. The 2011 WIMD incorporated multifaceted deprivation domains for small areas in Wales, enabling a comprehensive assessment of socioeconomic status. We explored both the WIMD score, which allowed exploring nonlinear associations, and its quintiles. The latter approach is commonly used in the assessment of health inequalities [3,59,60]. However, it involves information loss and other methodological problems [61]. Nonetheless, in our study, both approaches led to generally consistent findings. Finally, modelling by the WIMD 2011 individual domains, each in a separate model, has provided additional insights into the drivers of asthma inequalities, which should however be interpreted in the light of collinearity between some of these domains, particularly income, employment, education, and health.

Our study has some limitations. The WIMD 2011 is an area-level index [22], and therefore, caution is required when drawing person-level inferences. However, our findings are generally consistent with other studies in the UK and elsewhere which have found similar gaps of higher incidence of asthma symptoms and emergency hospitalisations and higher asthma severity in the more deprived areas [3,62–64]. Excluding patients with gaps in their primary care data from our cohort reduced the possibility of missing primary care data while insignificantly affecting the findings, as shown in the sensitivity analysis. However, this means people who died within the follow-up period, who might have more severe disease, were excluded from the main cohort in which health service utilisation was investigated. We did not exclude patients with diagnosis of chronic obstructive pulmonary disease (COPD), which may coexist with asthma, resulting in more severe symptoms and higher health service utilisation and mortality in the older ages [65–67]. The AMR formula is based on the number of prescriptions which did not necessarily reflect the actual prescribed dosage (puffs per day). However, data on actual dosage are currently not available in the primary care dataset that we have used. Residual confounders and mediators might have been involved in some of the observed associations in our study. Smoking is for example associated with socioeconomic deprivation [68]. The National Survey for Wales in 2016 to 2017 has found that adults in the most deprived quintile were 3 times more likely to smoke than those in the least deprived quintile [69]. Smoking is associated with a number of limiting long-term illnesses which were accounted for in the WIMD 2011 health domain. Smoking is also associated with poor asthma outcomes. Exposure to secondhand tobacco smoke among children with asthma is associated with reduction in pulmonary function and doubles the risk of hospitalisation for asthma exacerbation [70].

Therefore, differential smoking status might have partially mediated and confounded the observed association between socioeconomic deprivation and asthma-related health service utilisation, especially in secondary care. Asthma is also associated with a range of comorbidities, such as obesity and depression, which are associated with higher asthma severity and lower control and in which a socioeconomic gradient has been observed [71,72]. However, we did not control for the potential confounding effect of comorbidities.

## Implication for research, clinical practice, and public policy

The socioeconomic disparities that we have found in asthma-related health service utilisation highlight the need for multifaceted service improvement. Strategies are needed to aid optimal prescribing and prevent the excessive use of reliever inhalers [73]. In addition, the most deprived groups may require more effective health education on asthma self-management, including inhaler technique, adherence to treatment, and avoidance of triggers. However, those interventions alone are unlikely to bridge the socioeconomic gap in asthma outcomes. Rather, structural and social determinants, including the circumstances in which people are born and live, play a crucial role in asthma outcomes [74]. As with health inequalities in general, inequalities in asthma should ultimately be addressed by achieving equitable wider societal determinants of health, particularly educational opportunities, housing, and health service resourcing [75].

Avoidable health inequalities, in addition to being unfair, potentially waste resources. Given the high prevalence of asthma in Wales, even the modest gap in asthma health service utilisation, especially in secondary care, would result in avoidable, significant disease costs at the country level. To better understand and therefore tackle asthma inequalities, further research is needed to identify the most significant and modifiable determinants, estimate their avoidable financial cost to the public sector, and identify the most cost-effective service and public health interventions to reduce these inequalities and their burden.

## Conclusions

In conclusion, we have found consistent socioeconomic variations in asthma health service utilisation, prescribing, and death in Wales across all stages of patient care. Patients in the most deprived areas had poorer prescribing and were over 1.5 times more likely to both be urgently admitted to hospital and to die due to asthma compared with the least deprived areas. These inequalities are associated with avoidable harm and deaths to patients and costs to Wales. There is a pressing need to develop targeted service interventions and much wider societal policies to tackle such health inequalities.

## Supporting information

**S1 Checklist. Reporting checklists.**
(DOCX)

**S1 Text. Study design and data sources.**
(PDF)

**S2 Text. Patient selection.**
(PDF)

**S3 Text. Sensitivity analysis.**
(PDF)

**S1 Table. Clinical code sets.**
(PDF)

**S1 Fig. Distribution of the WIMD 2011 score and its quintiles in the study cohort.** WIMD, Welsh Index of Multiple Deprivation.
(PDF)

**S2 Fig. Model fit for the NB generalised linear regression models.** NB, negative binomial.
(PDF)

**S3 Fig. Distribution of AMR across the WIMD 2011 quintiles.** AMR, asthma medication ratio; WIMD, Welsh Index of Multiple Deprivation.
(PDF)

**S4 Fig. Proportion of emergency-to-total admissions in each quintile of the WIMD 2011 in the study cohort.** WIMD, Welsh Index of Multiple Deprivation.
(PDF)

## Acknowledgments

The authors would like to thank Prof Mike Gravenor, Dr Chris Newby, and Ms Rowena Bailey for their helpful feedback on the statistical aspects of this paper.

SER is part funded by The National Institute for Health Research Applied Research Collaboration North West Coast (NIHR ARC NWC).

The views expressed in this paper are those of the authors and not necessarily those of the NHS, the NIHR, or the Department of Health and Social Care.

## Author Contributions

**Conceptualization:** Mohammad A. Alsallakh, Sarah E. Rodgers, Ronan A. Lyons, Aziz Sheikh, Gwyneth A. Davies.

**Data curation:** Mohammad A. Alsallakh.

**Formal analysis:** Mohammad A. Alsallakh.

**Funding acquisition:** Sarah E. Rodgers, Ronan A. Lyons, Aziz Sheikh, Gwyneth A. Davies.

**Investigation:** Mohammad A. Alsallakh.

**Methodology:** Mohammad A. Alsallakh, Sarah E. Rodgers, Ronan A. Lyons, Aziz Sheikh, Gwyneth A. Davies.

**Project administration:** Mohammad A. Alsallakh, Sarah E. Rodgers.

**Software:** Mohammad A. Alsallakh.

**Supervision:** Sarah E. Rodgers, Ronan A. Lyons, Aziz Sheikh, Gwyneth A. Davies.

**Validation:** Mohammad A. Alsallakh.

**Visualization:** Mohammad A. Alsallakh.

**Writing – original draft:** Mohammad A. Alsallakh.

**Writing – review & editing:** Mohammad A. Alsallakh, Sarah E. Rodgers, Ronan A. Lyons, Aziz Sheikh, Gwyneth A. Davies.

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
