## [Editor Report · Decision Letter 0]

7 May 2020

Dear Dr Al Sallakh, 

Thank you for submitting your manuscript entitled "Impact of socioeconomic deprivation on asthma care and outcomes in Wales: a five-year national linked primary and secondary care cohort study" for consideration by PLOS Medicine.

Your manuscript has now been evaluated by the PLOS Medicine editorial staff and I am writing to let you know that we would like to send your submission out for external peer review.

Kind regards,

Caitlin Moyer, Ph.D.,

Associate Editor

PLOS Medicine

---

## [Decision Letter · Decision Letter 1]

2 Jul 2020

Dear Dr. Al Sallakh,

Thank you very much for submitting your manuscript "Impact of socioeconomic deprivation on asthma care and outcomes in Wales: a five-year national linked primary and secondary care cohort study" (PMEDICINE-D-20-01743R1) for consideration at PLOS Medicine. 

[LINK]

In light of these reviews, I am afraid that we will not be able to accept the manuscript for publication in the journal in its current form, but we would like to consider a revised version that addresses the reviewers' and editors' comments. Obviously we cannot make any decision about publication until we have seen the revised manuscript and your response, and we plan to seek re-review by one or more of the reviewers. 

We expect to receive your revised manuscript by Jul 23 2020 11:59PM. Please email us (plosmedicine@plos.org) if you have any questions or concerns.

We look forward to receiving your revised manuscript. 

Sincerely,

Emma Veitch, PhD

PLOS Medicine

On behalf of Clare Stone, PhD, Acting Chief Editor,

PLOS Medicine

plosmedicine.org

*As noted by one reviewer, it would be important to be careful around use of causal language given the study design, and there is not always consistency throughout the paper.

*Please structure your abstract using the PLOS Medicine headings (Background, Methods and Findings, Conclusions - "Methods and Findings" should be a single subsection header).

*In the last sentence of the Abstract Methods and Findings section, please include a brief summary of any key limitation(s) of the study's methodology.

*If possible, please reformat the intext reference callouts into PLOS Medicine style (sequential numerals in square brackets) - if using referencing software this should be fairly straight forward. Many thanks

*At this stage, we ask that you include a short, non-technical Author Summary of your research to make findings accessible to a wide audience that includes both scientists and non-scientists. The Author Summary should immediately follow the Abstract in your revised manuscript. This text is subject to editorial change and should be distinct from the scientific abstract. Please see our author guidelines for more information: https://journals.plos.org/plosmedicine/s/revising-your-manuscript#loc-author-summary

*Did your study have a prospective protocol or analysis plan? Please state this (either way) early in the Methods section.

a) Please clarify whether the analytical approach followed in this paper corresponded to one laid out in a prospective analysis plan (from your funding proposal, IRB or other ethics committee submission, study protocol, or other planning document written before analyzing the data) was used in designing the study, please include the relevant prospectively written document with your revised manuscript as a Supporting Information file to be published alongside your study, and cite it in the Methods section. A legend for this file should be included at the end of your manuscript. 

*The academic editor (AE) has some comments on one of the concerns of reviewer 3, please do take into account the view of the academic editor, this point should certainly be responded to by the authors in their rebuttal letter although as noted by the AE, the exact analysis proposed by the reviewer does not have to be added in revision. 

Comments from the academic editor:

I think reviewer 3's biggest issue around confounding individual/household wealth and other factors such as smoking vs geographic deprivation is an important one but it is simply not something that can be addressed with this sort of data linkage study. It would require linkage to other datasets (e.g census data) and that seems out of scope for this study.

Other key points:

1. The effect sizes are really very small for primary care utilisation and asthma reviews and although statistically significant I am not sure they translate to anything meaningful. Also for primary care and deaths, there does not seem much difference across quintiles 1-4 and only a difference with quintile 5. This suggests the association is not that clear when using categorical measures of deprivation and relates to reviewer 1's point about looking at it by numerical index.

2. I agree with reviewer 2 on many points and won't repeat them here except to emphasise the issue of defining medication use and converting to beclomethasone equivalents would make for more robust analyses. On Table 2, despite higher use of ICS in most deprived regions overall, the AMR is lower which perplexes me a little. This suggests it is a dosing/ adherence issue rather than prescribing in the first place.

3. Although the authors did a sensitivity analysis on people with non-continuous follow-up, I would have liked to see a sensitivity analysis that was not predicated on having at least one asthma prescription. There are a lot of people from Figure 2 who are excluded on this criterion. This group (non-prescribed) is likely heterogeneous and could influence the results in either direction - they have mild asthma and therefore don't use any services (primary care or hospital) or they have more severe asthma and don't access any treatment and get worse outcomes.

4. I think all primary results should be gender disaggregated given the findings from Table 4 and 5. I would be keen for more understanding to explain the much higher female utilisation of ED and hospitalisation overall. How does this relate to the current literature on the topic. Also it should be stated in the limitations that females were over-represented in primary health care datasets (this is a well-known limitation).

5. The authors have not done any age sub-group analyses and I would be keen to understand to what extent childhood vs adult asthma is driving the utilisation patterns observed.

6. The discussion suggests that more needs to be done in primary care for the most deprived groups but I think there needs to be much greater emphasis on social determinants of health and going outside the health care system to reduce inequities. I am fairly sceptical that improving primary health care would do much to address the disparities identified in this paper.

Comments from the reviewers:

Reviewer #1: I confine my remarks to statistical aspects of this paper. 

I have several issues to resolve before I can recommend publication.

First, in many places, the authors use causal langage (e.g. "impact") but this is not warranted. Words such as "associated" should be used instead.

Second, using quantiles of the wealth index is not good. In *Regression Modelling Strategies* Frank Harrell lists 11 problems with this and sums up "nothing could be more disastrous". Leave the index as a number and use splines to investigate nonlinearities. This will change some of the graphs (e.g. you can use density plots of index). 

Line 104ff I think that a beta regression would be better here. Logistic regression is used when the outcome is categorical. Here, it is continuous but bounded.

line 127 Why not use interactions with sex?

Peter Flom

Reviewer #2: This is an interesting study showing that socioeconomic deprivation is associated with worse asthma outcomes in Wales. There are however some improvements that can be made to the manuscript.

Comments:

1) The AMR was calculated by counting the number of ICS, ICS-LABA and SABA prescriptions. While this approach has the virtue of simplicity a drawback is that inhalers with different strengths or different numbers of doses are counted the same. Can the authors justify why they took this approach rather than counting the number of doses, or better still by calculating the total quantity of inhaled steroid used (in beclomethasone dipropionate equivalent units) and comparing this to the number of reliever doses?

2) Some of the data in Table 2 appear to be incorrect eg. Numbers of SABA, ICS and ICS-LABA inhalers seem too large and do not tally with the text. Also the number of A&E visits does not seem to tally between Table 2 and the text (line 185). Please could the authors carefully check all data in the text and tables to ensure they are correct.

3) A number of variables in Table 2 are heavily skewed to the right (eg. Length of stay, A&E visits, hospitalisations) so should not be expressed as mean (SD).

4) It is surprising that approximately 30% of hospital admissions were deemed as 'elective'. Elective admissions for asthma are very rare and I would expect 99% of admissions to be emergencies. This may be a coding issue and I would suggest simply presenting data on hospital admissions in aggregate.

5) It should be pointed out in the discussion that, while possibly reaching statistical significance, the difference in GP visits and asthma reviews between the lowest and highest deprivation quintiles is negligible and probably not of clinical significance.

6) Line 285-6: The authors mention that they did not perform sub-analyses for age groups - however, they have the data to do this and it would be useful to see if they could replicate the rather counter-intuitive results seen in England by Gupta et al.

7) What does the IRR in Line 199 refer to? I am unclear what incidence is being referred to in the context of the length of stay analysis (see also in Table 3).

8) The manuscript is generally well written, but there were a few typos I picked up: Line 56: Should read "Figure 1 illustrates..."; Line 88: There appear to be some missing words.; Line 116: Should read "For each count outcome variable..."

Reviewer #3: This cohort study explored the impacts of socioeconomic deprivation on asthma-related care and outcomes across primary and secondary care and mortality in Wales. 

While the study tackles an important research question, I have several major concerns about the adopted study design and the interpretation of study findings, as detailed below for the authors.

1. A major weakness of the study is that the analysis does not control for individual-/household-level socioeconomic status (SES). Consequently, the area-based measure of socioeconomic deprivation is most likely confounded by individual-/household SES, and the interpretation of the study's findings is less clear from that perspective. For example, a IRR of 1.58 for emergency admissions for asthma is observed in the most deprived quintile vs. least deprived quintile. This would appear to be an important association, except for the fact that individual-/household SES most likely accounts for much of this association. In that event, what is the value/implication of identifying a IRR of 1.58?

2. It is unclear why the outcome of asthma-related death is used. This refers to any mention of asthma in the death record, but not necessarily asthma as the underlying cause of death. Such a lack of specificity is a disadvantage in my opinion.

3. Some description is needed about the relevance of the area units used to measure socioeconomic deprivation (LSOAs). In addition, the authors state that they used the address with the longest duration during the follow-up period (2013-2017). What assumptions are being made about the timing of exposure in the LSOA in relation to the asthma outcomes -- is such a short-term effect really plausible and supported empirically in previous studies? 

4. The authors cite smoking as a potential residual confounder. However, it seems infeasible that smoking (as a confounder) would cause area-level socioeconomic deprivation. What other residual confounders were unable to be controlled for in the analysis that can be commented on in the Discussion?

5. The study cohort consisted of individuals with ever-diagnosed, continuously-treated asthma. If those who are socioeconomically deprived are less likely to be diagnosed and/or continuously treated for asthma, how might that have affected the findings of the study and their interpretation?

6. For the negative binomial model that was applied, please clarify what offset was used, and please indicate any evidence that the NB model was more appropriate than a zero-inflated NB model.

7. The lack of control for individual-/household-level SES and the implications of this should also be discussed in the Discussion section within the study limitations paragraph.

[LINK]

---

## [Decision Letter · Decision Letter 2]

17 Nov 2020

Dear Dr. Al Sallakh,

Thank you very much for re-submitting your manuscript "Association of socioeconomic deprivation with asthma care, outcomes, and deaths in Wales: a five-year national linked primary and secondary care cohort study" (PMEDICINE-D-20-01743R2) for review by PLOS Medicine.

I have discussed the paper with my colleagues and the academic editor and it was also seen again by three reviewers. I am pleased to say that provided the remaining editorial and production issues are dealt with we are planning to accept the paper for publication in the journal.

[LINK]

We look forward to receiving the revised manuscript by Nov 24 2020 11:59PM. 

Sincerely,

Caitlin Moyer,

Associate Editor 

PLOS Medicine

plosmedicine.org

Requests from Editors:

1.Competing interests:Please add this statement to the manuscript's Competing Interests: "AS is an Academic Editor on PLOS Medicine's editorial board."

Data availability statement: Please revise to include the accession number or search term needed to identify the correct dataset relevant for applications to access SAIL data.

Abstract: Background: The term “patient journey” is slightly unclear. We suggest revising to “across all levels of care; or across all stages of patient care” or similar. (Please also clarify at the first sentence of the Discussion, and the Conclusion paragraph of the Discussion).

Abstract: Methods and Findings: Here, you use the term “pseudonymised” but in the Methods section you use the term “anonymised” (page 7). Please choose the most appropriate term and use consistently throughout.

2. Abstract: Methods and Findings: Please provide the p values in addition to the 95% CIs for results presented for least vs. most deprived patient comparisons for asthma reviews, primary care consultations, controller to total asthma medications, and asthma related accidents, emergency attendances, and emergency admissions, hospital status and death (if p values are p<0.001 please note that, otherwise please provide the exact p value).

3. Abstract: Methods and findings: If possible please clarify the statement of limitations, such as “...and the potential for residual confounders and mediators.” or similar, depending on what is meant.

4. Abstract: Methods and findings: Please emphasize that the differences in primary care utilization and asthma reviews were significantly different, but were small in absolute size.

Abstract: Conclusions: Please revise the first sentence to: “In this study, we observed that the most deprived asthma patients…” or similar.

5. Abstract: Conclusions: In line with the above comment, please remove the mention of primary care utilization and asthma review from the last sentence of the conclusion, as the impact of changing these is not clear (as mentioned in the discussion).

6. Author summary: Why was this study done? Please combine the first two bullet points: We suggest: “Income, education, and region of living are known to affect a person’s health, and studies around the world have found links between asthma and these socioeconomic factors.”

7. Author summary: What did the researchers do and find? Please combine the last two bullet points: We suggest: “In the most disadvantaged areas, people went more often to emergency departments

for asthma, and were 50% more likely to be admitted to hospital and die from asthma, and had a slightly worse balance of asthma medications, being three times more likely to take too many reliever inhalers, compared to people in the least disadvantaged areas.” or similar. Please also clarify “slightly worse balance of asthma medications” (e.g. had a lower controller to total ratio of asthma medications).

8. Author summary: What do these findings mean? Please combine the last two bullet points, and please temper these statements slightly to reflect what can be determined from your study. We suggest: 

“GP encouragement that people receive and take enough preventing medications and self-manage their asthma well regardless of background, and wider policies to provide equal educational opportunities across the society may help to make the socioeconomic gap in asthma less severe.” or similar

9. Throughout manuscript text: Please remove the spaces from within brackets for in text references: (e.g. [1,2,3] rather than [1, 2, 3].

10. Introduction: Please expand slightly on the Introduction, addressing past research and explain the need for and potential importance of your study.

11. Methods: Please specify the significance level used (e.g., P<0.05, two-sided) and the statistical test used to derive a p value.

12. Methods: Reporting and supporting reproduction: Please include the reference to the supporting information file containing the STROBE and RECORD checklists (S1_Checklist; S2_Checklist).

13. Methods: Study Planning: Did your study have a prospective protocol or analysis plan? Please state this (either way) early in the Methods section. Please either include a copy of the prospective plan as a supporting information file, or clarify which analyses were prespecified. Thank you for mentioning that GAMs were developed in response to peer-review.

14. Results: For the sections Asthma-related primary care consultations and reviews, Asthma prescriptions in primary care, Asthma-related A&E attendances, Asthma hospitalisations, and Asthma-related deaths, please present p values for the results described alongside the 95% CIs where applicable.

15. Results: Page 16 (and throughout): Instead of “the elderlies” we suggest “older adults” or similar.

16. Results: Page 17: Please provide p values associated with: “Patients in the most deprived areas were 3.6 times more likely to have 12 or more SABA inhaler per year (risk ratio = 3.6 [2.9, 3.3]).”

17. Results: Page 18: Please change “where” to “were” in the following: “Overall, rates where highest in the youngest patients and steeply decreased in older ages (Fig 4 and Fig 8).”

18. Discussion: Please ensure that the Discussion is organized as follows: a short, clear summary of the article's findings; what the study adds to existing research and where and why the results may differ from previous research; strengths and limitations of the study; implications and next steps for research, clinical practice, and/or public policy; one-paragraph conclusion. Specifically, please include paragraph or section discussing the strengths and limitations of the study.

19. Data sharing: Please remove the section “Data sharing” from the main text, and ensure that all information in the Data availability statement of the manuscript submission form is correct.

20. References: Please check the reference list formatting. For example, in reference 6, only 3 authors are listed before “et al” but the first six should be listed. Also, it seems like that should be punctuation between the journal name and the year of publication. Please use the "Vancouver" style for reference formatting, and see our website for other reference guidelines:

https://journals.plos.org/plosmedicine/s/submission-guidelines#loc-references

21. Table 3: Please present the actual p values for each comparison, unless p<0.001. If it is possible to number/order the quintiles and describe them in the legend, we suggest this to streamline the presentation of the column headers now reading “Most deprived, Next-most deprived…)

22. Table 4: Please round the p value for the within-quintile comparison among females for deaths with any mention of asthma.

23. Figure 7: Please note in the legend that the shaded areas represent 95% CIs, if that is the case.

24. Figure 8: Please place a label on the scale of the heat-map of partial effects (in addition to the description in the legend).

25. Supporting information Figure S6: The legend for the bottom panel graph appears to be cut off.

26. Checklist: Please include the completed STROBE checklist and the RECORD checklist as Supporting Information files. When completing the checklists, please use section and paragraph numbers, rather than page numbers. Please add the following statement, or similar, to the Methods: "This study is reported as per the Strengthening the Reporting of Observational Studies in Epidemiology (STROBE) and The REporting of studies Conducted using Observational Routinely-collected health Data (RECORD) guidelineS (S1 Checklist; S2 Checklist)."

Comments from Reviewers:

Reviewer #1: The authors have addressed my concerns and I now recommend publication.

Peter Flom

Reviewer #2: Thank you, I am happy that my comments have been satisfactorily addressed by the authors.

Reviewer #3: I appreciate the authors' responses to my comments.

[LINK]

---

## [Editor Report · Decision Letter 3]

15 Jan 2021

Dear Dr. Al Sallakh,

I am writing concerning your manuscript submitted to PLOS Medicine, entitled “Association of socioeconomic deprivation with asthma care, outcomes, and deaths in Wales: a five-year national linked primary and secondary care cohort study.”

We have now completed our final technical checks and have approved your submission for publication. You will shortly receive a letter of formal acceptance from the editor.

Kind regards,

PLOS Medicine